**Data availability statement:** All relevant data are within the paper.

**Funding:** The author(s) received no specific funding for this work.

**Competing interests:** The authors have declared that no competing interests exist.

# Scalable and rapid nearest neighbor particle search using adaptive disk sector

**Jong-Hyun Kim**[1], **Jung Lee**[2]*

**1** College of Software and Convergence (Department of Design Technology), Inha University, Michuhol-gu, Incheon, South Korea, **2** Department of Computer Engineering, Hanbat National University, Yuseong-gu, Daejeon, South Korea

* airjung@hanbat.ac.kr

## Abstract

In this paper, we propose a framework for efficiently accelerating Nearest Neighbor Particle (NNP) calculations in a movable particle-based system by leveraging the dynamic changes in disk sectors. The NNP region based on particles and disk sectors is determined by the following three conditions: 1) The position of the disk resides within the range of neighbor particles. 2) The position of a neighbor particle exists within a disk sector. 3) A neighbor particle exists between the two vectors that form the disk sector. When all of these conditions are satisfied, we assume that there is a particle within the disk sector. In this paper, we automatically update the inspection range of NNP, which is the disk sector, based on the movement of particles. To calculate the dynamic changes in the disk sector, we control the direction, length, and angle of the disk based on the positions and velocities of particles. Ultimately, we accelerate the computation of NNP by utilizing the particles located within the calculated disk sector. The proposed acceleration method can be implemented simply, as it operates on the particles within the disk sector using closed-form expressions, without the explicit data structures like trees. Especially in the case of movable particles, unlike the conventional adaptive tree approach that requires continuous data structure updates, the proposed method can be efficiently utilized in applications requiring NNP. This is because it rapidly calculates collision areas using closed-form expressions that are adjusted according to the particles' motion. Our method yielded results that were 2 to 20 times faster compared to Hash tables or $K$-d trees in experiments conducted across diverse scenes. Furthermore, its scalability was demonstrated through its application in various scenarios (particle-based fluids, splash and foam, isoline tracking, turbulent flow, collision handling).

## Introduction

Particle-based systems find applications in a wide range of fields, including physics-based simulations, rendering, gaming, and visual special effects [1,2]. Unlike rigid or deformable bodies based on triangular meshes, particle-based systems face the challenge of efficiently handling collision detection for a large number of target entities. To address this, various optimization methods such as quadtree/octree [3], $K$-d tree [4], and hash tables [5] are

commonly employed. However, these acceleration techniques can be inefficient from a performance perspective because they require data structure updates whenever the positions of particles change. In situations where there are not a large number of entities to perform collision detection on, the process of creating and updating acceleration data structures, as mentioned earlier, can sometimes take longer to execute than the actual collision detection operations themselves. This makes them less suitable for applications that require real-time performance, such as games or virtual reality content. In this paper, we propose a novel data structure for accelerating NNP searches in 2D space by dynamically altering disk sectors based on the positions and velocities of particles.

Particle-based physics simulations are valuable for developing and validating mathematical approaches in discrete environments. They also find utility in the field of computer graphics for representing various materials. Prominent examples of these approaches include Smoothed Particle Hydrodynamics (SPH) [6], Material Point Method (MPM) [7], Position-Based Dynamics (PBD) [8], and Position-Based Fluids (PBF) [9]. Both SPH and PBD calculate various physical forces (such as local fluid density and pressure) between particles and utilize kernel-based weighting functions for this purpose. In PBD and PBF, constraints are employed to model forces that maintain the rest length between particles, bending forces, and collision handling. The aforementioned approaches use iterative solvers to approximate solutions, which necessitates iterative access to neighbor particles. During this computation process, a significant portion of the time is spent collecting neighbor particles, and this collecting process accounts for a majority of the overall runtime performance.

This issue is evident in various applications within the field of computer graphics. One of the notable techniques is Photon Mapping, which involves emitting a large number of photon particles from a light source [10]. During the rendering process, nearest neighbor search is employed to approximate the illumination at a specific location, and in this process, $K$-nearest photons are required. Omitting the neighbor access process during rendering would lead to a significant degradation in image quality. Therefore, it is common practice to compute the energy for lighting using an approximation based on the $K$-nearest particles. As a result, a significant portion of the runtime performance of this technique relies on the computation of NNP for multiple iterations across all pixels in the image. To efficiently address this problem, various methods for hashing nearest neighbor photons have been proposed [11–13]. In this paper, our focus is on the adaptive disk sector approach, which allows for efficient computation of NNP. The domain we aim to address is not confined to fixed-point searches but instead encompasses dynamic scenes, such as SPH simulations, involving particles with continuously changing positions and velocities.

## Problem statement

Nearest Neighbor Problem (NNP$^\star$) is a fundamental problem extensively studied in computer science and mathematics. This operation involves finding a set of points that are closest to a given point from different sets of points and is frequently applied in fields such as data mining, pattern recognition, and machine learning.

NNP$^\star$ can be solved using a hash table based on Locality-Sensitive Hashing (LSH). LSH is a technique that maps similar data points to the same bucket with a high probability, making it useful for nearest neighbor search. Hash tables are well-suited for such operations because they have an intuitive implementation and do not require complex data structures like trees. However, its performance is heavily dependent on the chosen resolution, making it crucial to find the optimal value. In the worst-case scenario, memory usage can become inefficient, and

search performance may also degrade. The optimal value for hash table application can vary depending on the specific application, and it can be challenging to find the best value when the distances between points are not consistent.

To address this issue, many solutions utilize a multi-resolution approach, with Quadtree and $K$-d tree being prominent examples. Utilizing Quadtree for solving NNP$^\star$ offers the following advantages: 1) This technique is efficient in low-dimensional spaces. 2) In scenes with few and sparse points, space can be efficiently represented, leading to memory savings. However, despite these advantages, there are also the following drawbacks: 1) In high-dimensional spaces, trees can become unbalanced, and it may be necessary to search through many nodes. 2) When many points are concentrated in a few regions, the performance of a tree structure can significantly degrade. 3) This tree structure is designed to perform optimally in cases with uniform point density. Consequently, as point density becomes more uneven, its performance may degrade.

A data structure that addresses the limitations of Quadtree is the $K$-d tree, and using it to solve NNP$^\star$ offers the following advantages: 1) It is efficient in high-dimensional space. 2) Efficient traversal is enabled by the balanced structure of $K$-d trees, making them scalable even for large datasets. However, despite these advantages, there are also the following drawbacks to using $K$-d trees: 1) In cases with skewed data distributions, the tree structure can become imbalanced in some cases, potentially leading to performance degradation. 2) Datasets with uneven point density can result in significant memory usage. 3) In large datasets, the need to search through many nodes to find the nearest neighbor can lead to slower search performance.

The methods for solving NNP$^\star$ typically fall into one of the three categories mentioned earlier. In conclusion, to mitigate the limitations mentioned earlier, we propose a fast data structure similar to a hash table, without the need for tree structures. In this process, our approach optimizes using adaptive disk sectors rather than manually determining the resolution, as done in a hash table. Prominent spatial data structure techniques include hash tables and multi-resolution-based $K$-d trees, which have been extended into various algorithms such as $A^*$ [50] and $D^*$ algorithms [51], primitive trees [52], and primal trees [53]. In this study, we propose a novel approach to improving low-level data structures, making it applicable not only for NNP but also for shortest path and path-finding techniques that utilize these structures.

- Technical contributions
    1. We introduce a method that can rapidly define the search range using a disk sector-based approach based solely on particle velocity, without the need for explicit data structures like commonly used spatial data structures such as hash tables or $K$-d trees. While calculating anisotropic features typically requires computationally intensive processes like Singular Value Decomposition (SVD), in this study, we were able to quickly define the search range using only the particle's velocity and angle.
    2. We propose a method for calculating the search range for moving particles by introducing a kernel that can reliably update the disk sector based on particle velocity.
- Potential applications with our method
    1. Based on our method, we can rapidly construct NNPs in the direction of particle velocity and extend this approach to compute SPH simulations using an ellipsoidal kernel.
    2. We can extend the algorithm to rapidly construct NPPs in the direction of particle velocity and then determine the conditions under which diffuse particles should be generated and initiate their creation.

3. Our algorithm can be extended to construct NNPs in the direction of velocity, compute a particle-based level set, and then extract isolines using the marching cubes algorithm.

4. Strands such as hair or fur are composed of particles, and our algorithm can be extended to handle self-collision by constructing NNPs using the method proposed in this paper to identify candidate particles for collision handling.

5. Our algorithm can be extended to calculate turbulent flow in a particle-based system by constructing NNPs based on particle velocity, allowing for the generation and maintenance of turbulence.

## Related work

In this section, we will briefly explore methods for efficiently handling particle-based collision detection, a prominent application that utilizes NNP$^\star$.

Acceleration techniques based on space partitioning for efficiently handling collision detection with a large number of particles can be broadly categorized into two main categories: 1) Adaptive structures based on Bounding Volume Hierarchies (BVH), represented by Octrees, $K$-d trees, Binary Separating Planes (BSP) trees, Oriented Bounding Box (OBB) trees. 2) A hash table utilizing regular grid structure.

BVH is a common data structure that utilizes a hierarchical arrangement of bounding volumes to accelerate collision detection [14,15]. To perform intersection tests for arbitrary 3D models using this technique, collision detection is initially carried out with the top-level bounding volume that encloses the model. If a collision occurs at the bounding volume, the next step involves collision detection for the triangles composing the 3D model enclosed within the bounding volume.

Octree (or Quadtree in 2D) is a tree-like data structure based on BVH that employs a tree structure containing eight child octants [3,16]. It's referred to as an adaptive spatial partitioning structure because node splitting in Octree only occurs in regions where objects exist. Ultimately, the leaf nodes of the Octree are allocated to spaces that contain the actual triangles of objects (see Fig 1a).

In contrast to Octree, the $K$-d tree employs a structure where it divides space into two regions along $X$, $Y$, or $Z$ planes. Typically, $K$-d trees have fewer child nodes compared to Octrees, which results in smaller tree sizes. One of the characteristics of $K$-d trees is that they

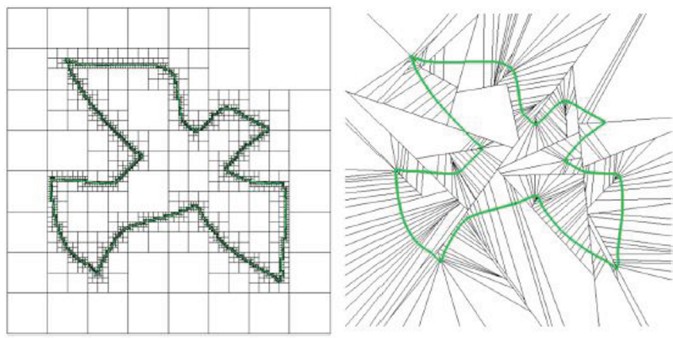

(a) Quadtree decomposition    (b) BSP tree decomposition

**Fig 1. Comparison between Quadtree and BSP Tree.**

https://doi.org/10.1371/journal.pcbi.0311163.g001

can provide more accurate NNP results through tree searches, particularly when the data distribution within the search area is uneven. Hash tables are more accurate and efficient for data distributions that are regular. However, $K$-d trees are preferred when dealing with irregular data distributions, making them a common choice for accelerating operations like ray-tracing [4,17,18].

BSP tree is a technique that divides space using arbitrary planes. As the name suggests, this technique is also a spatial partitioning structure and is commonly used not only for collision handling but also in the field of robotics (see Fig 1b).

OBB tree is a hierarchical fitting method that rotates bounding boxes close to the shape of an object [19,20]. Bounding boxes, during the hierarchical construction process, can overlap with each other, so they do not partition space.

A hash table utilizing a regular grid is a non-hierarchical structure that divides space into boxes of the same size, unlike the approaches described earlier [5,21]. While this technique performs quickly, in complex scenes or with high resolutions, it can demand a significant amount of memory and may become computationally inefficient.

In addition, collision detection using spatial partitioning approaches can be broadly categorized into two domains: 3D object space and 2D image space, depending on the nature of the domain being examined. Object space approaches typically calculate collision detection using the topology and geometry information of objects. Image space approaches project the information of 3D objects into 2D image space, resulting in reduced computational requirements and faster collision processing.

Recently, approaches that utilize hardware acceleration have been proposed. Knott and Pai used graphic hardware to detect collisions between polyhedra [22]. In this technique, they detected collisions occurring between polyhedra using a virtual ray-casting operation. Govindaraju et al. proposed an algorithm that can be efficiently utilized in deformable and fracture simulation scenarios where collision processing is relatively frequent [23]. Greß and Zachmann performed intersection calculations in collision detection of polygonal models using graphic hardware-based methods [24]. This approach involves processing the tree traversal stages that occur in BVH on the GPU in a single pass, enabling fast collision detection. Govindaraju et al. accelerated collision detection between objects by computing visibility queries on the GPU [25].

Sigg et al. leveraged hardware acceleration to accelerate the computation of the signed distance field (SDF) [26]. They computed SDFs quickly by utilizing a limited region, known as the local distance field, when calculating SDFs on the GPU. Kipfer et al. proposed the row sorting technique to simulate the movement of particles and approximate collisions between particles [27]. They constructed scenes using a memory compression technique called height field, but this approach was not sufficient for representing complex 3D models. Kolb et al. efficiently handled collision processing in a large number of particles using a depth map [28]. Hoff et al. constructed SDFs based on Voronoi features and performed fast collision processing by handling this process on graphic hardware [29]. Chen et al. presented a technique that quickly detects collisions between deformable objects using graphic hardware [30]. Govindaraju et al. introduced a new culling algorithm based on graphic hardware to reduce the number of collision tests between models [31]. Wong and Baciu proposed a framework for rapidly computing collision tests in deformable simulations using GPUs [32]. They extended this framework to enable real-time collision processing through GPU-based continuous collision detection [33].

There are also specially designed methods available for solving NNP$^\star$ more efficiently. As mentioned earlier, conventional tree-based approaches are designed with the aim of improving the construction phase to reduce lookup time. Dynamic memory allocation for inserting

all particles during the tree construction phase can be time-consuming, ultimately slowing down the performed operations. To prevent this, it is more efficient to allocate static memory before running the algorithm. Tree-based approaches generally use memory to establish neighborhood relationships between particles. Otair [34], Gieseke [35], Zhou [36], and Qui [37] proposed an approach for efficiently handling nearest neighbor queries using a $K$d-tree. Another drawback of using tree structures on GPUs is that the memory access pattern on GPUs is generally in a random access format, making it difficult to achieve optimal computational efficiency.

One of the efficient approaches that can be utilized for NNP$^{\star}$ is using a grid structure to improve nearest neighbor search. 1) Green employs a virtual grid to hash particles into specific grid cells [38]. In this approach, once the particle insertion phase is completed, all particles are reordered using radix sort to improve coherent memory accesses. Sorted grid cells are used to find potentially neighboring particles. This technique can suffer from performance degradation when dealing with a large number of particles due to the runtime limitations of the radix sort algorithm. 2) The approach proposed by Hoetzlein improved upon Green's method by utilizing atomic functions [39].

Kawada et al. proposed a method that adds 'bounding-space' around each grid cell to eliminate the need for additional lookup tables [40]. However, during the process of searching for particles belonging to multiple boundary regions, there can be issues with particle duplication. Therefore, an additional step for removing duplicate particles is required, leading to increased memory usage.

Garcia et al. proposed an optimization method for the $K$-nearest neighbor (KNN) search problem utilizing CUDA and CUBLAS APIs [41]. They calculated the distance between query points and reference points using CUDA kernels, although the primary algorithm, beyond acceleration by GPU threads, is similar to conventional hash tables. Xueyi Wang introduced the kMkNN ($k$-Mean for $k$-Nearest Neighbors) algorithm, which employs $K$-means clustering and the triangle inequality to swiftly locate the NNP in high-dimensional spaces [42]. This method unfolds in two stages: the buildup state, where a simple $K$-means clustering is used for preprocessing the training dataset instead of complex tree structures like $K$-d trees; and the searching stage, where, given a query object, kMkNN starts from the closest cluster to find the nearest training object, reducing distance computations through the triangle inequality. Liaw et al. presented a method for efficiently finding NNP in high-dimensional spaces, proposing the use of an Orthogonal Search Tree (OST) for rapid indexing of datasets [43]. This algorithm recursively partitions the dataset along coordinate axes to construct the OST. Each node in the tree represents a data partition, allowing for efficient pruning of the dataset by traversing the tree based on the distance between query points and partition boundaries. To enhance search efficiency, this method employs a branch-and-bound technique to terminate the search process early once KNNs are identified. However, OST's recursive partitioning approach results in high memory requirements, potentially leading to memory shortages, especially for large datasets.

Tobias Plötz and Stefan Roth proposed a method called Neural Nearest Neighbors Networks, which can determine processes requiring adjacent information, like in CNNs (Convolutional Neural Networks), through learning [54]. They demonstrated the effectiveness of their approach by applying it to applications such as image denoising and super-resolution. However, since it only works within block-designed, image-based networks, it is practically inapplicable to 3D particle simulations. More recently, Chen et al. proposed a nearest neighbor approximation technique based on a neural similarity metric [55]. As this data structure is designed for large-scale recommendation systems, it is only applicable in specific applications and is difficult to apply to moving particles.

## Proposed framework

### Collision detection between particles and disk sectors

In this paper, the flame is assumed to exhibit the characteristics of an incompressible multiphase fluids flow, and the modeling is based on this assumption. The Incompressible Navier-Stokes equations are expressed as equations of momentum conservation and mass conservation (see Eqs 1 and 2).

The method proposed in this paper consists of two execution phases: 1) It performs collision checks between static disk sectors and particles, and 2) dynamically updates the disk sectors based on the positions and velocities of the particles. This dynamic updating process automatically determines the size of the area to perform collision checks.

To model the first phase, similar to the SPH technique, we assume that particles have finite regions and perform three types of collision checks as illustrated in Fig 2. In this figure, the white particles represent the positions of disk sectors where collision checks are performed, and they serve as the pivot positions for these disk sectors. Additionally, the red and black particles represent neighboring particles. The red particles represent the particles that collide with the disk sector.

Fig 2a depicts a scene where collisions occur between the disk sector and the radius range of neighboring particles. The position of the red particle is not inside the disk sector, but its radius range contains the pivot position of the disk sector (see the case (C) in Fig 2a). Similar to most particle-based systems, we also assume that particles have a finite radius range for calculating physical quantities. In Fig 2a, all three cases, (A), (B), and (C), do not contain particles within the disk sector. However, the case (C) is considered a collision because the position of the disk sector falls within the radius range.

Most particles will be filtered out in the case shown in Fig 2a. However, in situations where particles have high velocities, like in Fig 2b, collisions may be detected because the positions of neighbor particles fall within the disk sector (see the cases (A) and (C) in Fig 2b).

Finally, Fig 2c is designed to more accurately detect cases similar to those in Fig 2a and 2b. Indeed, collisions occur exclusively within the disk sector. Therefore, the regions outside of the two vectors used to define the disk sector (see red line in Fig 2c) are excluded from the collision detection process (see the cases (A) and (B) in Fig 2b).

The detailed numerical techniques for implementing the three cases mentioned earlier are as follows :

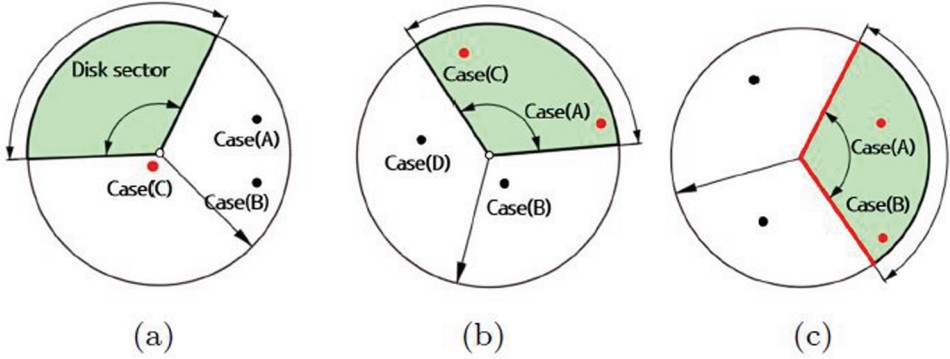

**Fig 2. Various collision types between disk sector and particles.**

https://doi.org/10.1371/journal.pcbi.0311163.g002

- The case when the pivot position of the disk sector is inside the radius range of a neighbor particle (see Fig 2a)

  This case can be computed using the implicit function of the 2D general equation of the sphere (see Eq 1).

$$\left(x_p - x_c\right)^2 + \left(y_p - y_c\right)^2 - r_p^2 < 0 \tag{1}$$

where $(x_p, y_p)$ represents the position of the neighbor particle, $(x_c, y_c)$ represents the pivot position of the disk sector, and $r_p$ is the radius representing the finite region of the particle. The radius represents the range for performing collision detection between the particle and the disk sector. Consequently, if the pivot position of the disk sector is within the particle's radius, it satisfies Eq 1, leading to the conclusion that a collision has occurred.

The second and third cases are calculated using the internal division of the two vectors that make up the disk sector.

- The case where the particle's position is inside the disk sector (see Fig 2b)
- The case where the particle's position lies between the two vectors composing the disk sector, and the distance from the particle's position to the disk sector is smaller than the sum of the particle's radius and the disk sector's length (see Fig 2c)

  If the above conditions are met, it can be said that the particle lies between the two vectors composing the disk sector, $\mathbf{v}_1$ and $\mathbf{v}_2$. In this paper, these conditions are calculated as follows (see Eq 2).

$$p - c = \alpha\mathbf{v}_1 + \beta\mathbf{v}_2 \tag{2}$$

where $p$ represents the position of the neighbor particle, which is $(x_p, y_p)$, and $c$ represents the pivot position of the disk sector. $\alpha$ and $\beta$ are positive weighting factors used when calculating the vector's internal division. Eq 2 utilizes vector internal division to interpolate between two vectors, $\mathbf{v}_1$ and $\mathbf{v}_2$, in a ratio of $\beta:\alpha$. To position a particle between the two vector spaces, both $\alpha$ and $\beta$ must be greater than or equal to 0, and the distance from the disk center to the particle must be smaller than the magnitude of $\mathbf{v}_1$. In all other cases, it is determined that the particle is outside the disk sector (see Eq 3).

$$\|p - c\| < \|\mathbf{v}_1\| \tag{3}$$

The $\mathbf{v}_1$ and $\mathbf{v}_2$ that make up the disk sector have the same size, so changing $\mathbf{v}_1$ to $\mathbf{v}_2$ in the above equation does not affect the result. Eqs 2 and 3 are not modeled based on an implicit function for the disk sector. However, in this paper, we designed the algorithm based on the two vectors that make up the disk sector because we only need to know the length to determine if a particle exists within the disk sector. As a result, when all the aforementioned conditions are satisfied, it is considered that the particle and the disk sector have collided.

## Disk sector update

In this paper, dynamic changes are introduced to the collision region, represented by the disk sector, to update its direction and size according to the movement of particles. Particle position and velocity were used to control the pivot position, angle, and length of the disk sector.

Fig 3 illustrates two scenarios based on the magnitude of the particle's velocity. First, in both cases, the direction of the disk sector is aligned with the particle's velocity. Fig 3a depicts a scenario where the particle's velocity is slow. In the case where the particle has a slow velocity, the disk sector's length is set to be short and the angle is made wide. This is because there is a higher likelihood of collisions occurring in the tangential direction rather than in the direction of the particle's motion. Fig 3b depicts a scenario where the particle's velocity is fast. In this case, the particle is moving quickly, so there is a higher likelihood of collisions occurring in the area where it passes through in the direction of its motion rather than in the tangential direction. Therefore, the disk sector's length is relatively long, and the angle is set to be small. As a result, when the velocity is slow, the length is set short, and the angle is set large, aiming for a search range that is close to isotropic. On the other hand, when the velocity is fast, the length is set long, and the angle is set small to reflect anisotropic characteristics in the search range.

To implement the aforementioned method, the position of the particle is set as the pivot position of the disk sector, denoted as $c$. Then, the length of the disk sector, represented as $l_{pv}$, is calculated (see Eq 4).

$$l_{pv} = \left\| v_{pv} \right\| \delta \tag{4}$$

where $v_{pv}$ represents the particle's velocity, and $\delta$ is a weight used to adjust the length of the disk sector. The angle of the disk sector, denoted as $a_{pv}$, is updated by interpolating between minimum and maximum values based on the magnitude of $v_{pv}$ (see Eqs 5–7).

$$a_{pv} = \left(1 - w^*\right) a_{pv}^{max} + w^* a_{pv}^{min} \tag{5}$$

$$w^* \left(w, h\right) = \begin{cases} 0, & 0 \leq h \\ \frac{\gamma + log_\gamma w}{\gamma}, & else \end{cases} \tag{6}$$

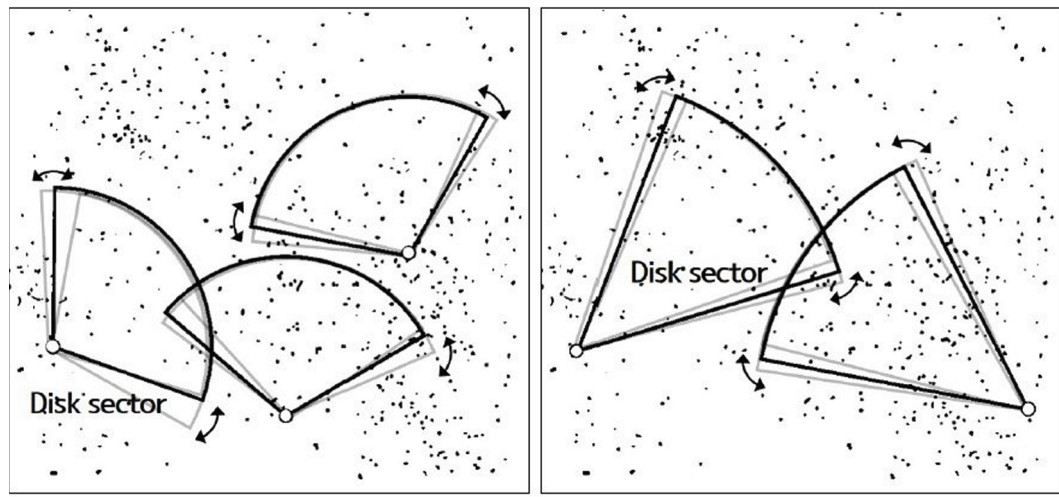

(a) Length : short, angle : large (b) Length : long, angle : small

**Fig 3. Area of the disk sector that varies with the particle velocity (gray disk: trajectory to update disk based on the particle's velocity and position).**

https://doi.org/10.1371/journal.pcbi.0311163.g003

$$w = \frac{min\left(max\left(l_{pv}, \left\|v_{pv}^{min}\right\|\right), \left\|v_{pv}^{max}\right\|\right)}{\left\|v_{pv}^{max} - v_{pv}^{min}\right\|} \tag{7}$$

where $w$ is a weight used for interpolating the angle, and $\gamma$ is the base of the *log* function, which is set to 3 in this paper. As a result, this function determines the angle of the disk sector, taking into account the particle's velocity. When the particle's velocity is too slow or too fast, the value of $w$ can change excessively, ultimately compromising the stability of the simulation. To mitigate this issue, we use a refined value of $w$, denoted as $w^*$, to interpolate the angle.

In this paper, we use a weight $w$ to automatically adjust the search range based on the velocity of moving particles. A parameter is required to define the search range per angle, and we interpolated the angle based on user-defined minimum and maximum values. However, if the velocity is too fast or too slow, the search range may excessively expand or contract, making it difficult to stably calculate NNPs. To address this issue, we introduced $w^*$, a refined weight based on a kernel, to determine the angle-based search range. As shown in Fig 4, $w^*$ converges to a kernel value of 1, preventing it from growing uncontrollably and allowing us to calculate NNPs more stably.

Fig 4 shows changes in $w^*$, demonstrating that using $w^*$ provides a more stable disk sector than using $w$, which increases linearly with the particle's velocity. In the equilibrium state region, where the velocity is slow, the disk sector's angle is not reduced, and it is designed to converge to $a_{pv}^{max}$ in the relatively short distance region to provide a larger disk sector area. The disk sector, which changes with the particle's velocity, plays a crucial role in collision detection. Observing the slope of $w^*$, we can see that it increases significantly as the particle's velocity starts to rise ($w$: 0.11–0.175). Consequently, the disk sector's angle increases rapidly. When the particle's velocity is relatively high, increasing the angle of the disk sector rapidly can cause the disk sector to become excessively large. To address this, the kernel design aims to attenuate the gradient of $w^*$ as the particle's velocity increases, ensuring that the disk sector does not grow too large ($w$: 0.675–0.825).

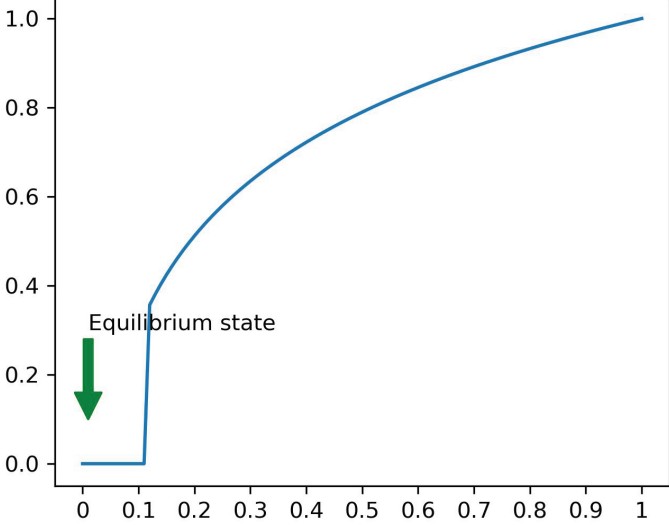

**Fig 4. The weight function inside the kernel $w^*$ (*X*-axis: $w$ (see Eq 7), *Y*-axis: weight value $w^*$).**

https://doi.org/10.1371/journal.pcbi.0311163.g004

## Experimental results

### Scene initialization

To analyze the disk sector-based dynamic collision region technique proposed in this paper from various perspectives, several scenarios were devised. The fundamental essence of the scenarios is as follows. It is assumed that the particles for NNP are not at fixed positions but rather move over time, similar to SPH. The computational workload increases because the acceleration data structure needs to be updated every time the particles move. We perform a comparison with existing techniques in a simulation environment with a relatively high computational workload to analyze the acceleration performance.

### Results

To demonstrate the superiority of our method, we conducted NNP calculations in a confined area with a large number of randomly placed particles using a simple and intuitive scenario (see Fig 5). In the 2D example shown in Fig 5, we randomly distributed 10,000 particles. The target for NNP calculation is the red dot, and green particles that exist within the user-specified radius range are considered as NNPs.

The simplest method to find the NNP is to perform a complete search over all particles. This approach involves searching over all particles regardless of the velocity of the target particle, making it computationally intensive and resulting in unnecessary calculations. Fig 5 illustrates the results of NNP calculations where only the target's position was considered, without accounting for its motion. The NNP calculations took an average of 0.003 s. Fig 5a and 5b show the results of calculating neighboring particles based on the moving target particle over time.

Fig 6 shows the results of NNP calculations performed using our method. This was tested in the same environment as shown in Fig 5, and our method took an average of about $160\mu s$ during the NNP calculation process. Fig 6a demonstrates the results of disk sector-based NNP calculations in a scenario where particles have fast velocity, while Fig 6b shows the results for a scenario with relatively slower velocities. The final computed NNP, represented by the green particles, consists of the candidate particles located within the disk sector that are also

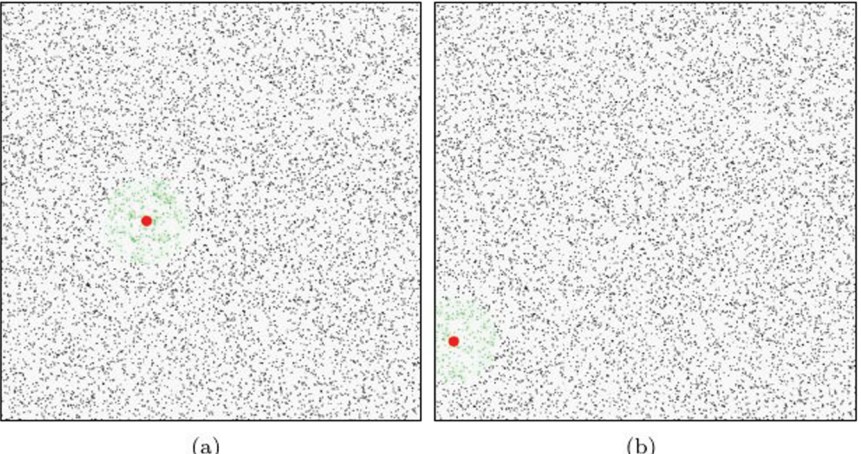

(a)                               (b)

**Fig 5. Nearest neighbor search with brute-force search (green particle: NNP, red dot: target particle).**

https://doi.org/10.1371/journal.pcbi.0311163.g005

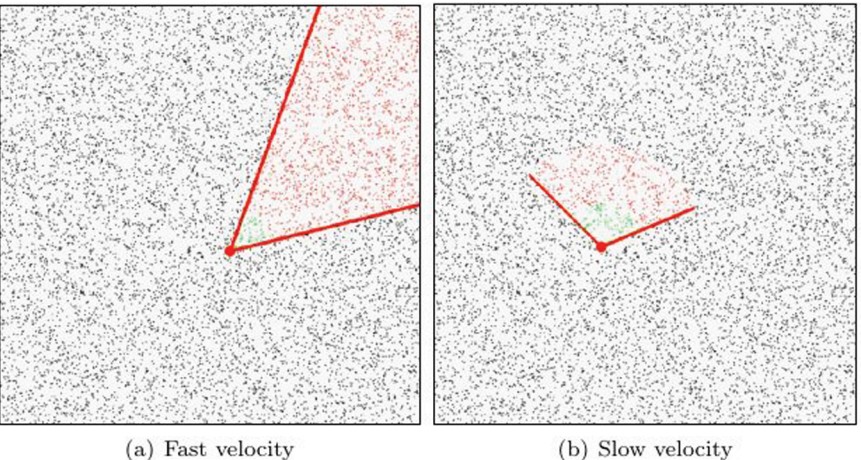

**Fig 6. Nearest neighbor search with our method (small red particle: candidate particle, green particle: NNP, red dot: target particle).**

https://doi.org/10.1371/journal.pcbi.0311163.g006

within the user-specified radius range. In Fig 5, all particles were considered for collision checks, whereas in Fig 6, only the small red particles were involved in the collision checks. This significantly reduced the number of particles to be examined, resulting in a performance improvement of approximately 18.7 times.

Fig 7 shows the results of NNP tests conducted on particles that are distributed according to various densities, unlike the regular distribution scenario discussed earlier. Our method automatically adapts the disk area based on the direction and magnitude of particle velocity, and it efficiently computes NNP without an explicit acceleration data structure (see Fig 7).

Generally, hash tables determine their resolution based on the spacing of particles, so they may not be suitable for solving NNP$^\star$ in environments where particle density is uneven. Fig 8 shows the results of calculating NNP using a hash table. The hash table algorithm's performance can vary depending on the grid resolution, and it often requires manual tuning.

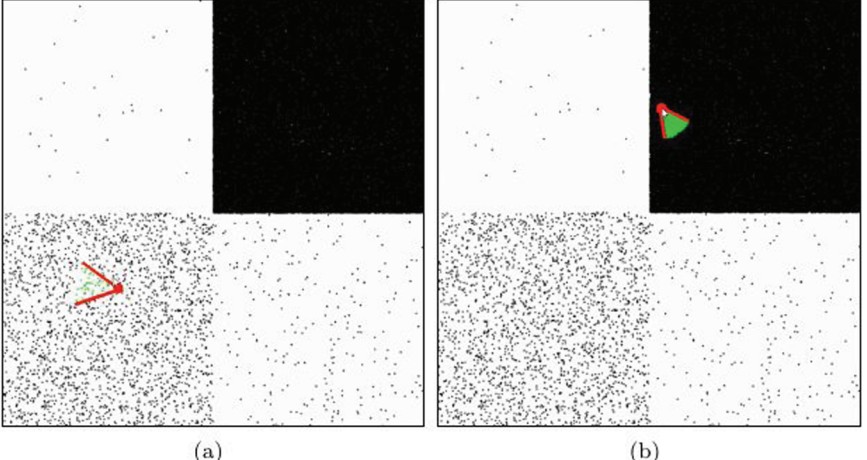

**Fig 7. Nearest neighbor search in various particle density with our method.**

https://doi.org/10.1371/journal.pcbi.0311163.g007

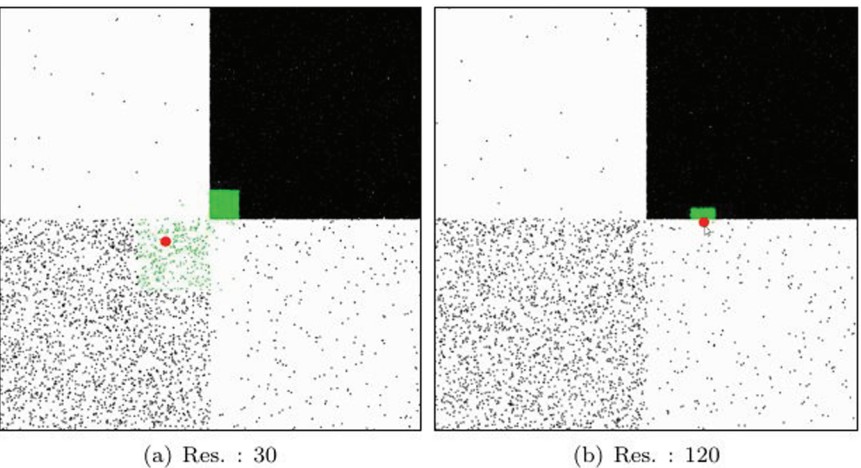

**Fig 8. Nearest neighbor search in various particle density with Hash table.**

https://doi.org/10.1371/journal.pcbi.0311163.g008

Tuning can be challenging, especially when dealing with scenes of varying resolutions. In cases where the grid resolution is low, it takes a significant amount of time to construct the Hash table. Conversely, when the resolution is high, the performance deteriorates due to the increased number of NNPs. This challenge makes it difficult to create an optimized table in scenes with non-uniform particle density (see Fig 8).

Utilizing a $K$-d tree can alleviate the issues mentioned earlier due to its multi-resolution property. However, in high-density areas, constructing the tree requires more computational effort, and the balancing of the tree can vary, leading to potentially significant differences in the depth at which NNPs are searched (see Fig 10). For movable particles, the tree must be updated every frame, and during this process, the issue of tree imbalance can lead to inefficient traversal of the tree when searching for the nearest neighbors.

Fig 9 illustrates the number of NNPs obtained using our method and previous approaches. Our method is efficient as it can rapidly calculate NNPs without the need to update data structures in dynamic environments. The number of neighbor particles varies depending on the particle's velocity, which affects the search range. In the divided quadrants (LT: Left top, RT: Right top, LB: Left bottom, RB: Right bottom), during the 10th frame shown in the Figure, the NNP calculations were performed in the LB quadrant where particles were relatively more distributed. From frames 50 to 230, the NNP was computed in the LT quadrant, which had a relatively sparse particle distribution. The reason for the flickering pattern in this area, despite a uniform distribution of neighbor particles, is that the velocity of the target particle affects the search range. From frame 240 to 320, NNP computations were performed while moving from the LB to the RB and then to the RT. Similarly, even in the densely distributed RT, the number of neighboring particles can vary based on the target particle's velocity.

Fig 9b shows the results of calculating NNP based on a 30×30 resolution hash table. We can observe that the number of neighbor particles varies significantly despite having the same experimental environment as shown in Fig 9a. In the early stages, from 0 to 10 frames, NNP was calculated in the LB. From 30 to 40 frames, it was calculated in the RB, and the largest computational load was in the RT for NNP calculations. Unlike our method, the computational load of the hash table depends on the grid resolution, and this characteristic is clearly

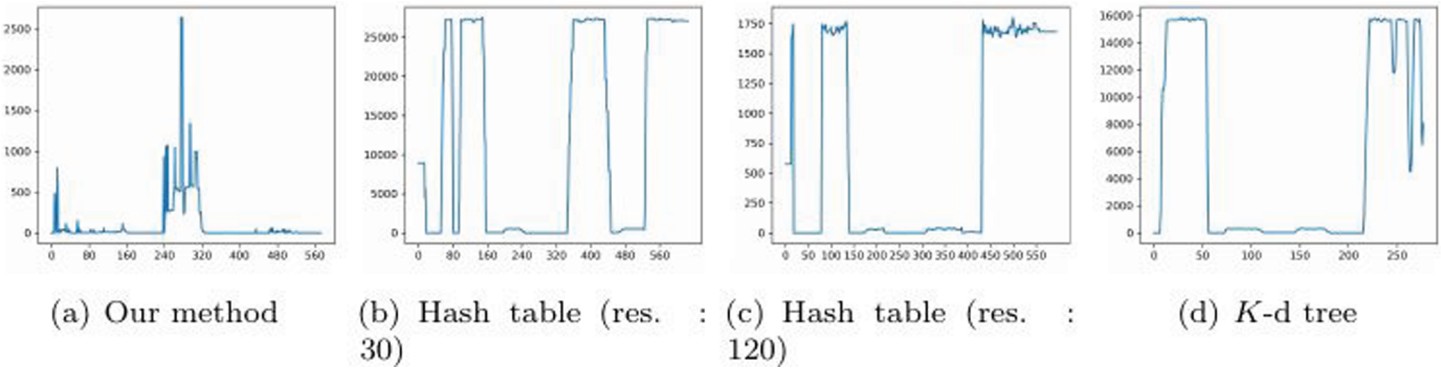

**Fig 9. The number of NNPs generated per frame for different methods (*X*-axis: frame #, *Y*-axis: number of NNPs).**

https://doi.org/10.1371/journal.pcbi.0311163.g009

reflected in the number of NNP counts as well. In our method, the number of neighbor particles varies with the velocity of target particles, leading to fluctuation patterns. In contrast, in the hash table, the number of neighbor particles is determined based on particle density, resulting in a more consistent count within the same quadrant. The hash table relies solely on grid resolution and does not take into account the motion of target particles. Consequently, it often identifies a large number of neighbor particles that may not be relevant to the movement of the target particles. In the experimental results, it is evident that the number of particles fluctuates with particle density (see Fig 9b). This problem is not resolved by increasing the grid resolution. A similar pattern was observed even in a 120×120 resolution hash table (see Fig 9c). Hash tables find varying number of NNPs depending on the resolution, but constructing a high-resolution hash table takes longer computation time. Moreover, this technique is limited in that it only calculates NNP within predefined bucket sizes and doesn't consider particle motion.

Fig 9d shows the results of NNP calculations using the *K*-d tree-based approach. The pattern of NNP counts, in general, exhibited a similar shape to that of the hash table. In addition to regular grids, most multi-resolution data structures collected NNP counts similar to the hash table. This technique also depends solely on the shape of the data structure independently of particle motion, resulting in the collection of almost the same number of particles. This pattern is clearly evident in the results. Our method is not based on a fixed data structure. Rather, it takes into account particle motion, which can affect both particle density and their physical movements. Therefore, in scenarios like particle simulations where particle positions change every frame, there is no need for the hashing and tree update processes, making the implementation straightforward and efficient.

Fig 11 visualizes the computational time from the update phase to finding the NNP based on the data structures of hash table and *K*-d tree. Hash tables and *K*-d trees rely solely on particle density for constructing their data structures, which impacts not only the number of NNP but also the computational time and memory usage. Fig 11 shows the computation time for the scene used in Fig 10, experimented with hash tables and *K*-d trees, represented with rainbow colors. Both the hash table and *K*-d tree methods show blue colors when the number of particles is small, indicating that NNP can be computed quickly. However, when the number of particles is large, the search time increases for both hash tables and *K*-d trees, resulting in relatively longer computation times. Particularly at boundary regions, hash tables exhibit more pronounced aliasing effects compared to *K*-d trees. This occurs because the *K*-d tree supports multi-resolution as a spatial data structure, leading to smoother color transitions

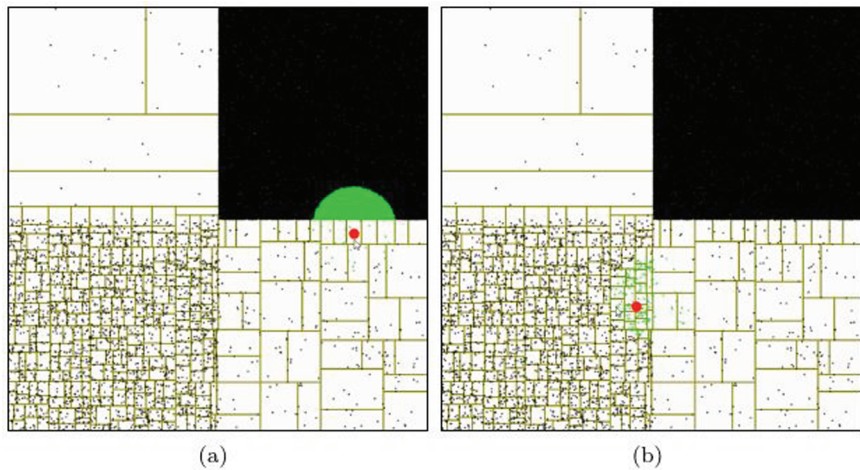

**Fig 10. Nearest neighbor search in various particle density with *K*-d tree.**

https://doi.org/10.1371/journal.pcbi.0311163.g010

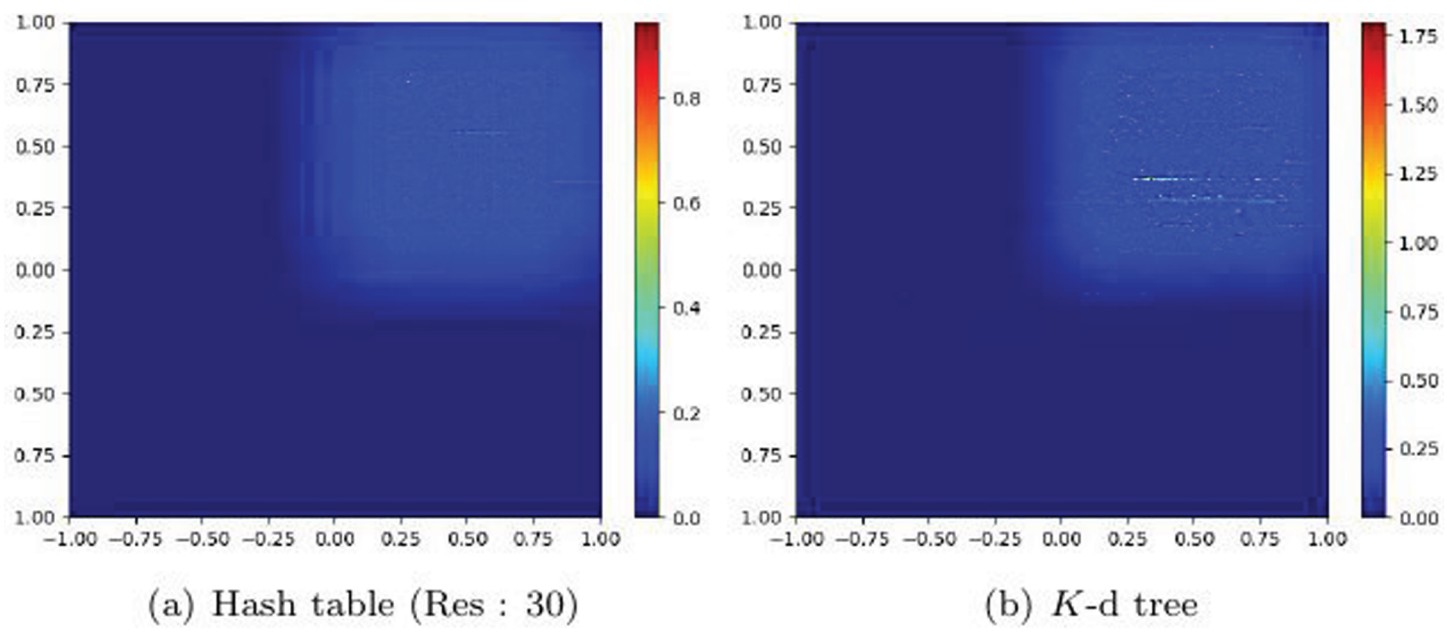

**Fig 11. Visualization of the time spent performing a nearest neighbor search.**

https://doi.org/10.1371/journal.pcbi.0311163.g011

even near boundaries with significant density differences. In contrast, the hash table displays greater variations in search time near these boundaries due to density differences, which is visualized as aliasing. The time differences within dense regions for the *K*-d tree are primarily due to differences in search performance depending on tree depth, whereas the hash table, which has relatively constant-time complexity, shows less variation in computation time.

## Comparison experiments between sparse and dense regions

In this section, we present the results of NNP experiments in both sparse and dense regions. In this experiment, the user can freely move the mouse cursor, and the algorithm considers

the cursor's position as the target position for NNP calculation. In this experiment, the results of the well-utilized acceleration data structures, $K$-d trees and hash tables, were compared with our method, using the collision check environment as depicted in Fig 5.

Fig 12a and 12b presents the results of experimenting with finding NNP using $K$-d trees with different bucket sizes. In this technique, as the bucket size decreases, the tree depth increases. Nodes extracted as collision candidates through the $K$-d tree are marked with red boxes. The final NNPs are collected using only the particles included in those nodes. The green particles shown in Fig 12a represent the final NNPs, and the time taken to find them is $3368\mu s$. When performing NNP checks on 500,000 particles, it took 0.13s (see Fig 13). As the number of particles increases, the NNP checking time increases because the $K$-d tree needs to be updated each time. In the scene with a larger bucket size in Fig 12b, NNP checking took $2,049\,\mu s$ for 10,000 particles and 0.09 s for 500,000 particles. There was only a slight difference depending on the bucket size. Fig 12c and 12d shows the results of NNP detection using a hash table. The NNP experiments were conducted with various resolutions when constructing the hash table, and the respective times taken were $362\,\mu s$ (res.: 30) and $621\,\mu s$ (res.: 120). For the case with 500,000 particles, the respective times taken were 0.015 s (res.: 30) and 0.017 s (res.: 120). Our method exhibited the best performance with $120\,\mu s$ for sparse and 0.008 s for dense scenarios. Table 1 summarizes the performance comparison results based on these experiments.

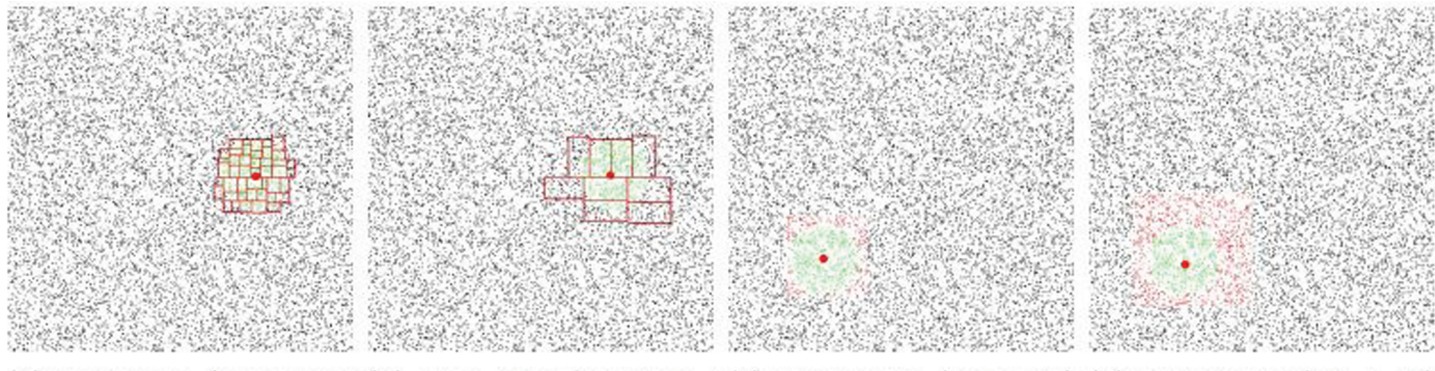

(a) $K$-d tree (red box : leaf node, bucket size : 10)   (b) $K$-d tree (red box : leaf node, bucket size : 100)   (c) Hash table (res. : 30)   (d) Hash table (res. : 15)

**Fig 12. Comparative results of collision detection using $K$-d trees and hash tables (red: candidate particle, green: NNP, red sphere: target position).**

https://doi.org/10.1371/journal.pcbi.0311163.g012

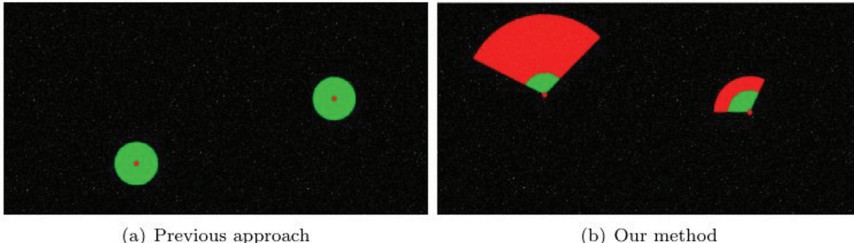

(a) Previous approach                          (b) Our method

**Fig 13. Comparison between the previous method (brute-force search) and our method (red: candidate particle, green: NNP, red sphere: target position).**

https://doi.org/10.1371/journal.pcbi.0311163.g013

**Table 1. The average computation time comparison between the previous method and our method.**

|  | Fig 12 | Fig 13 |
|---|---|---|
| Hash table (res.: 15) | 310$\mu s$ | 0.015s |
| Hash table (res.: 30) | 362$\mu s$ | 0.016s |
| Hash table (res.: 120) | 621$\mu s$ | 0.017s |
| $K$-d tree (bucket size: 10) | 3,368$\mu s$ | 0.13s |
| $K$-d tree (bucket size: 100) | 2,049$\mu s$ | 0.08s |
| Our method | 120$\mu s$ | 0.008s |
| Hash table (res.: 15)/Ours | ×2.5 | ×1.8 |
| Hash table (res.: 30)/Ours | ×3 | ×2 |
| $K$-d tree (bucket size: 10)/Ours | ×28 | ×16.2 |
| $K$-d tree (bucket size: 100)/Ours | ×17 | ×10 |

In this experiment, NNP calculations are performed from a pre-defined circular position, rather than a position moved in real-time by the user, to examine the performance. The scenario is configured as shown in Fig 16, where the target position for NNP is specified in a circular shape. In this process, 10,000 particles and 500,000 particles were used for coarse and dense scenarios, respectively. As evident from the experimental results, our method demonstrated the fastest NNP execution time compared to the previous approaches. These results suggest that our method, while not relying on the explicit creation and update of a specific data structure for calculating NNP within an isotropic range, efficiently identifies particles based solely on the velocity of the target particle, forming an adaptive disk sector. In the next section, we explore the applicability of the NNP detected using the adaptive disk sector to various applications, in addition to the performance validation shown earlier, and analyze the results.

Fig 14 is a chart that records the computation time frame by frame for the scene in Fig 16a. While Table 2 presents the average computation time, this measure does not reveal the variations between frames. Therefore, this paper aims to analyze our method further through additional data. To avoid a monotonous distribution of particles in a regular pattern, this paper employs random sampling for their placement.

In Fig 14, the target position for calculating the NNP is represented by a pink particle. Since the target position is iteratively visited frame by frame to calculate the NNP, the variation in computation time was relatively significant compared to other scenes. Our method deals with moving particles, necessitating the consideration of velocity, which we

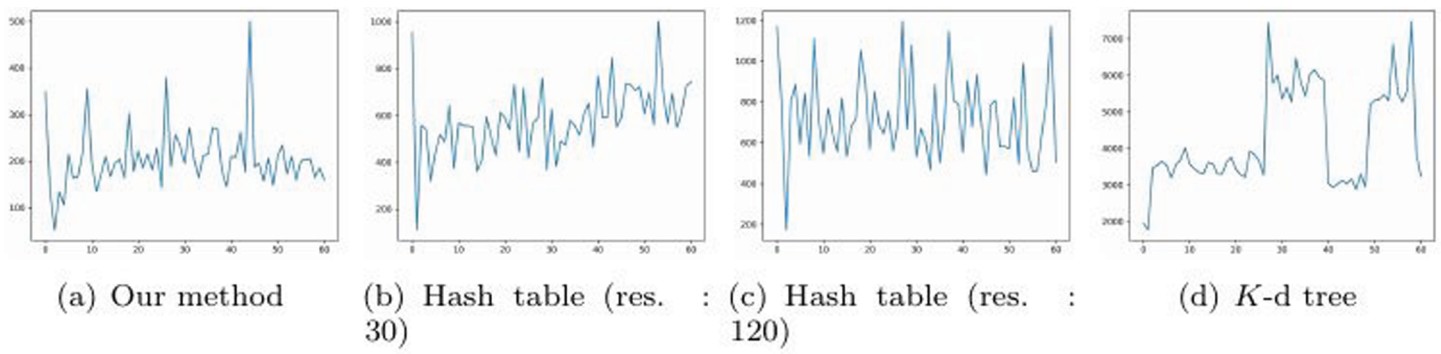

**Fig 14. The processing time ($\mu s$) for each method of NNP as shown in Fig 16a (*X*-axis: frame #, *Y*-axis: computation time).**

https://doi.org/10.1371/journal.pcbi.0311163.g014

**Table 2. The average computation time comparison between the previous method and our method.**

|  | Fig 16a | Fig 16b |
| --- | --- | --- |
| Hash table (res.: 30) | 581.2$\mu s$ | 13,882.3$\mu s$ |
| Hash table (res.: 120) | 722.6$\mu s$ | 16,885.6$\mu s$ |
| $K$-d tree (bucket size: 10) | 4,279.6$\mu s$ | 127,097.4$\mu s$ |
| $K$-d tree (bucket size: 100) | 2,466.3$\mu s$ | 90,300.6$\mu s$ |
| Our method | 205.6$\mu s$ | 5,039.3$\mu s$ |
| Hash table (res.: 30)/Ours | ×2.8 | ×2.7 |
| Hash table (res.: 120)/Ours | ×3.5 | ×3.4 |
| $K$-d tree (bucket size: 10)/Ours | ×20.8 | ×25.2 |
| $K$-d tree (bucket size: 100)/Ours | ×11.9 | ×17.7 |

approximated through the difference between frames as: $\left(\frac{p_{curr}-p_{prev}}{\Delta t}\right)$. In Fig 14a, because the velocity is not generated through mouse drag or simulation, it appears similar to uniform motion, resulting in minimal variation in computation time due to the similarity in search range within our method. This characteristic was similarly observed in the hash table but, as examined in Table 2, our method was the fastest. As mentioned earlier, the multiresolution $K$-d tree, which varies its division criteria based on the distribution of particles, showed less variation than the hash table but had a longer computation time than our method.

Although the chart in Fig 14 shows significant variation in computation time, the measurement unit is microseconds($\mu s$), so the perceived impact is minimal. Fig 15 presents results from experiments with denser particles, showing patterns mostly similar to those in Fig 14. The relatively high computation time in the first frame of Figs 14a and 15a is due to the lack of velocity at the start position, resulting in a default larger search range. In subsequent frames, there is a tendency to converge towards optimized performance aligned with velocity changes. Additionally, the initial frames in Figs 14b and 15b show increased computation time due to the inclusion of the time to construct the hash table. Since the $K$-d tree construction time is relatively fast, the computation time in the first frame was not significant.

## Friedman test to validate

In this paper, we validated the results of Table 2 using the Friedman test. The analysis conducted using the Friedman test is presented in Table 3. Typically, a p-value of 0.05 is used as a

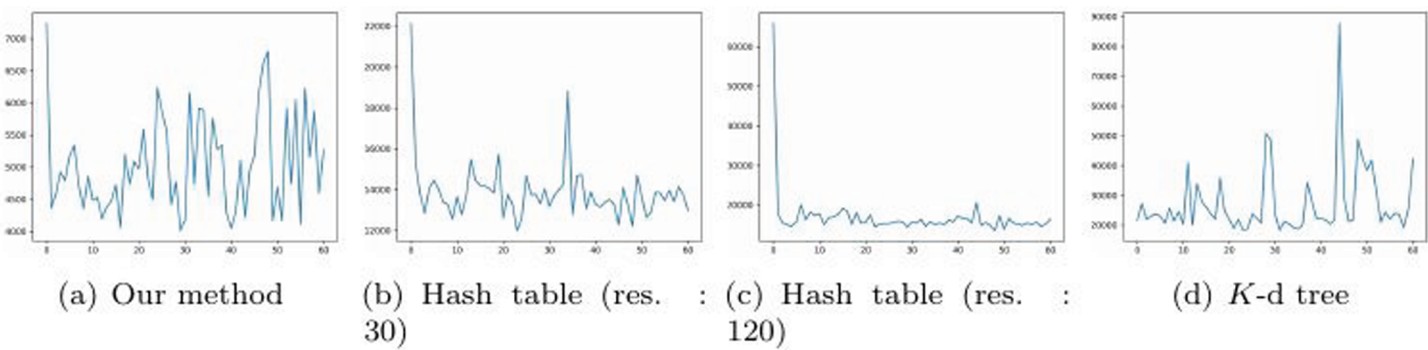

**Fig 15. The processing time ($\mu s$) for each method of NNP as shown in Fig 16b ($X$-axis: frame #, $Y$-axis: computation time).**

https://doi.org/10.1371/journal.pcbi.0311163.g015

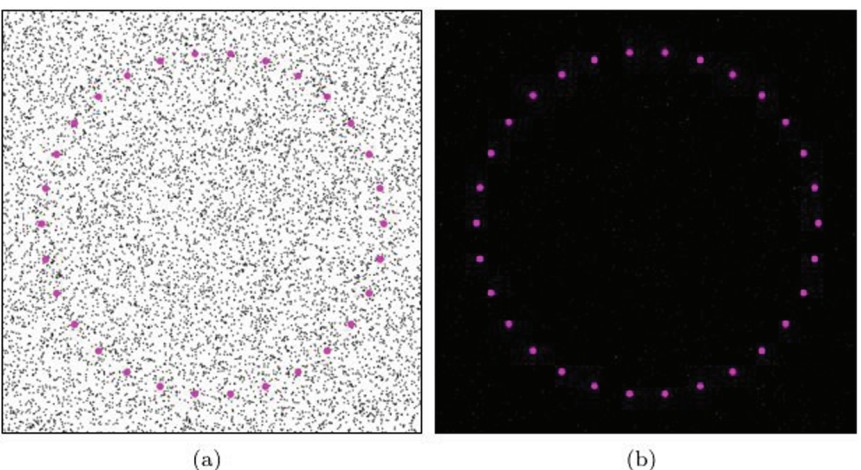

**Fig 16. NNP test scene with predefined circular position.**

https://doi.org/10.1371/journal.pcbi.0311163.g016

**Table 3. Table of analysis using the Friedman Test (k: number of scenes, n: number of methods).**

| Method | Fig 16a | Fig 16b | Fig 16a Ranks | Fig 16b Ranks |
|---|---|---|---|---|
| Hash table (res.: 30) | 581.2$\mu s$ | 13,882.3$\mu s$ | 1 | 2 |
| Hash table (res.: 120) | 722.6$\mu s$ | 16,885.6$\mu s$ | 1 | 2 |
| $K$-d tree (bucket size: 10) | 4,279.6$\mu s$ | 127,097.4$\mu s$ | 1 | 2 |
| $K$-d tree (bucket size: 100) | 2,466.3$\mu s$ | 90,300.6$\mu s$ | 1 | 2 |
| Our method | 205.6$\mu s$ | 5,039.3$\mu s$ | 1 | 2 |
| Sum of Ranks | | | 5 | 10 |
| k | 2 | | | |
| n | 5 | | | |
| Q | 5 | | | |
| p-value | 0.02535 | | | |

threshold, with lower values indicating stronger significance of the data. The p-value obtained from our analysis was approximately 0.025, which, despite the relatively small sample size of the entire experimental group, suggests satisfactory results when considered alongside the actual computation times.

## Applications

In this section, we examine several areas where our method can be applied to discuss its scalability and versatility.

**Particle-based fluids using ellipsoidal kernels.** Various kernel methods have been proposed to improve the viscous fluid motion in Smoothed Particle Hydrodynamics (SPH) fluids. We apply our method to a new SPH simulation technique that is modeled using ellipsoidal kernels instead of spherical kernels [44]. This approach deforms the kernel into an ellipsoidal shape based on the velocity vector of the particle. The key idea is to deform the isotropic kernel, for example, assuming that the kernel is elongated in the $X$-axis direction, we can express it using the ellipsoid equation as Eq 8.

$$\left(\frac{x}{a}\right)^2 + \left(\frac{y}{b}\right)^2 + \left(\frac{z}{c}\right)^2 = 1 \qquad (8)$$

where we assume that $a > b = c$ and normalize such that $abc = 1$, indicating that the kernel is elongated in the $X$-axis direction. This kernel reduces the interaction forces (e.g., viscous effects, pressure forces) between smoothing particles in the direction of the shorter axis, rather than applying the same force in all directions. If we assume that the length of the major axis of the ellipsoid is equal to the displacement distance of the particle during the time step, it can be expressed as Eq 9.

$$2\sqrt{a^2 - b^2} = |\mathbf{u}| \Delta t \qquad (9)$$

And this equation can be rewritten with respect to $a$ (see Eq 10).

$$a = f(k) = \sqrt{\frac{k + \sqrt{k^2 + 4}}{2}} \qquad (10)$$

where is defined as follows: $k = \left(\frac{1}{2}|\mathbf{u}|\Delta t\right)^2$. For very fast particles, they can interact with particles that are far away, and this can negatively impact numerical stability. To address this, $f(x)$ is refined for these particles.

Each particle can form an oriented ellipsoidal kernel based on its velocity, and this deformed kernel can be interpreted as a scaled, stretched, and rotated version of the spherical kernel. Therefore, the transformation from kernel (local) coordinates to world coordinates and its inverse transformation are as Eqs 11 and 12.

$$\mathbf{r} = \mathbf{RSz} \qquad (11)$$

$$\mathbf{z} = \mathbf{S}^{-1}\mathbf{R}^{-1}\mathbf{r} \qquad (12)$$

where $\mathbf{z}$ and $\mathbf{r}$ represent the kernel and world coordinate vectors, respectively. $\mathbf{S}$ is the scaling matrix, and $\mathbf{R}$ is the rotation matrix for rotating in the $\frac{\mathbf{u}}{|\mathbf{u}|}$ direction.

These matrices can be computed using the particle's velocity vector, $\mathbf{u}$. The scaling matrix, $\mathbf{S}$, is a diagonal matrix where each diagonal component is the denominator of the ellipsoid equation (see $a, b, c$ in Eq 8).

The SPH kernel can be expressed using transformation matrix as Eq 13.

$$W_e(\mathbf{r}, h) = W(\mathbf{z}, h) \qquad (13)$$

where $W_e$ represents the new ellipsoidal kernel, and $W$ is the spherical basis kernel. With the ellipsoidal kernel, the gradient operator can be expressed as Eq 14.

$$\nabla W_e(\mathbf{r}, h) = \frac{d\mathbf{z}}{d\mathbf{r}} \cdot \frac{\partial W(\mathbf{r}, h)}{\partial \mathbf{z}} \qquad (14)$$

As mentioned earlier, these approaches derived the ellipsoid equation to create an ellipsoidal-shaped kernel. Furthermore, it calculates anisotropy based on velocity rather than the distribution of neighbor regions. Unlike the method proposed by Yu et al., this study employs velocity for anisotropic deformation, a process typically computed using Singular Value Decomposition (SVD) [49]. During this process, the set of neighbor particles, denoted by $j$, is computed within the adaptive disk sector, $\Omega$, calculated using our method. SPH has

the characteristic of being regularly distributed, so when applying our method to this technique, we constrained the maximum length of the $l_{pv}$ to the radius used in SPH, which is $2h$. When particles move very slowly, the computation of nearest particles occurs in almost isotropic ranges, showing results similar to traditional methods. As particles accelerate, the computation involves the calculation of adaptive disk sectors, eliminating the explicit need for computing anisotropic kernels. Our method, characterized by an anisotropic search range, can be effectively applied to an ellipsoidal kernel-based SPH solver (see Fig 17), providing a stable representation of SPH fluids using ellipsoidal kernels. In Jo et al.'s approach, an isotropic kernel is computed and then deformed in the velocity direction, causing stability to decrease as the speed increases [44]. In contrast, our method considers particle velocity in determining the search range, leading to improved stability.

**Diffusing particles in particle-based fluids.** Ihmsen et al. proposed a method for representing spray, foam, and bubbles in particle-based fluids [45]. This method represents secondary effects by calculating potential energy in neighbor particles to express diffused material. In this paper, the neighbor particle set $p_j$ for water particles is calculated within the adaptive disk sector $\Omega$. The detailed process for applying this is as follows.

The potential energy for trapped air is calculated as follows: $I_{ta} = \Phi\left(v_i^{diff}, \tau_{ta}^{min}, \tau_{ta}^{max}\right)$. Here, $\Phi$ represents the clamping function proposed by Ihmsen et al. [45], and $v_i^{diff}$ is calculated as Eq 15.

$$v_i^{diff} = \sum_{i \in \Omega} \left\|\mathbf{v}_{ij}\right\| \left(1 - \hat{\mathbf{v}}_{ij} \cdot \hat{\mathbf{x}}_{ij}\right) W\left(\mathbf{x}_{ij}, h\right) \tag{15}$$

where $W$ is a radially symmetric weighting function, and $\hat{\mathbf{v}}_{ij}$ is the normalized relative velocity calculated between particles $p_i$ and $p_j$. $\hat{\mathbf{x}}_{ij}$ is the normalized distance vector.

The potential energy for the wave crest is calculated as follows: $I_{wc} = \Phi\left(\tilde{\kappa}_i \cdot \delta_i^{vn}, \tau_{wc}^{min}, \tau_{wc}^{max}\right)$. The term $\delta_i^{vn}$ represents a weighting factor indicating whether the particle is moving in the

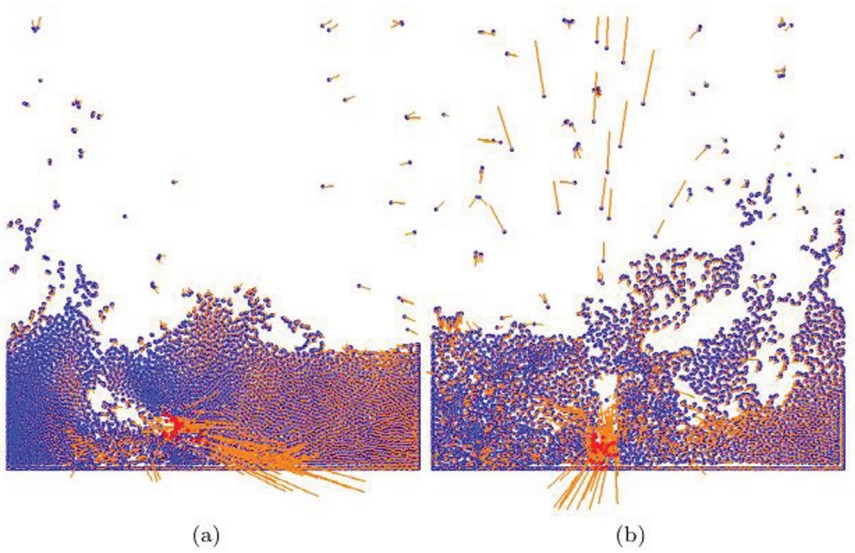

(a)                                          (b)

**Fig 17. The integration of our method into the SPH solver based on the ellipsoidal kernel (blue particle: water particle, red particle: active particles manipulated by the mouse, orange line: velocity of water particle).**

https://doi.org/10.1371/journal.pcbi.0311163.g017

normal direction, and $\tilde{\kappa}_i$ is calculated as Eq 16.

$$\tilde{\kappa}_i = \sum_{j \in \Omega} \tilde{\kappa}_{ij} \tag{16}$$

$$\tilde{\kappa}_{ij} = \begin{cases} 0 & \hat{\mathbf{x}}_{ij} \cdot \hat{\mathbf{n}}_i \geq 0 \\ \kappa_{ij} & \hat{\mathbf{x}}_{ij} \cdot \hat{\mathbf{n}}_i < 0 \end{cases} \tag{17}$$

where The term $\hat{\mathbf{n}}_i$ represents the normalized surface normal, and $\kappa$ is the surface curvature calculated from the water particles (see Eq 18).

$$\kappa_i = \sum_{i \in \Omega} \kappa_{ij} = \sum_{j \in \Omega} \left(1 - \hat{\mathbf{n}}_i \cdot \hat{\mathbf{n}}_j\right) W\left(\mathbf{x}_{ij}, h\right) \tag{18}$$

When sampling diffuse particles, they are generated uniformly within a cylinder, similar to the previous technique. The number of generated diffuse particles is calculated as follows: $n_d = I_k \left(k_{ta} I_{ta} + k_{wc} I_{wc}\right) \Delta t$. $I_k$ is the clamping function used by Ihmsen et al. [45], and $k_{ta}$ and $k_{wc}$ represent the number of particles generated per frame due to trapped air and wave crest, respectively. These values can be adjusted by the user.

When advecting diffuse particles, the process involves calculating the average local fluid velocity, $\tilde{\mathbf{v}}_f$, using the adaptive disk sector $\Omega$. The remaining steps are the same as in the previous technique [45].

$$\tilde{\mathbf{v}}_f\left(\mathbf{x}_d, t + \Delta t\right) = \frac{\sum_{f \in \Omega} \mathbf{v}_f\left(t + \Delta t\right) W\left(\mathbf{x}_d\left(t\right) - \mathbf{x}_f\left(t\right), h\right)}{\sum_{f \in \Omega} W\left(\mathbf{x}_d\left(t\right) - \mathbf{x}_f\left(t\right), h\right)} \tag{19}$$

where $W$ is a normalized symmetric kernel, and $\mathbf{v}_f$ is as follows: $\mathbf{v}_f\left(t + \Delta t\right) = \frac{\mathbf{x}_f(t+\Delta t) - \mathbf{x}_f(t)}{\Delta t}$.

Fig 18 shows the diffuse particles generated by incorporating our method. Not only are splash effects naturally and smoothly represented in the areas where water particles collide and create turbulence, but the simulation also accurately captures bubble and foam effects within the water (see Fig 19). This scene effectively demonstrates the representation of secondary effects through user interaction. As mentioned earlier, our method is not only applicable to directional ellipsoidal kernels but also integrated seamlessly with the generation and advection of secondary effects based on the direction of water particles.

**Isoline tracking in particle-based fluids.** Our method can also be utilized for tracking fluid surfaces. However, it should be applied in a different manner than the two methods mentioned earlier. Typically, in particle-based fluids, the surface is approximated based on a density-weighted average function rather than being generated in the direction of high momentum. Therefore, we configure the disk sector bidirectionally. We calculate $p_j$ within the positive/negative direction of the region $\Omega$, with the pivot position $c$ as the reference, in the disk sector (see Eq 20).

$$\phi\left(\mathbf{x}\right) = \sum_{j \in \Omega^{\pm}} \frac{m_j}{\rho_j} W\left(\mathbf{x} - \mathbf{x}_j, h\right) \tag{20}$$

where $\Omega^{\pm}$ represents the adaptive disk sector indicating the bidirectional region. Fig 20 shows the isolines of fluid surfaces extracted using Marching Squares. Particle-based level-set approximates the level-set based on particle density rather than the nearest distance from

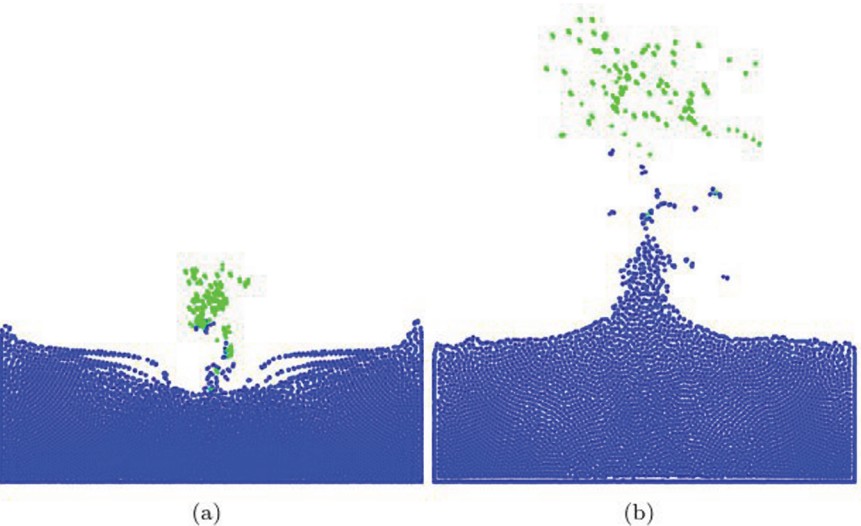

**Fig 18. Results of integrating our method into an secondary effects solver (blue particle: water particle, green particle: diffuse particle).**

https://doi.org/10.1371/journal.pcbi.0311163.g018

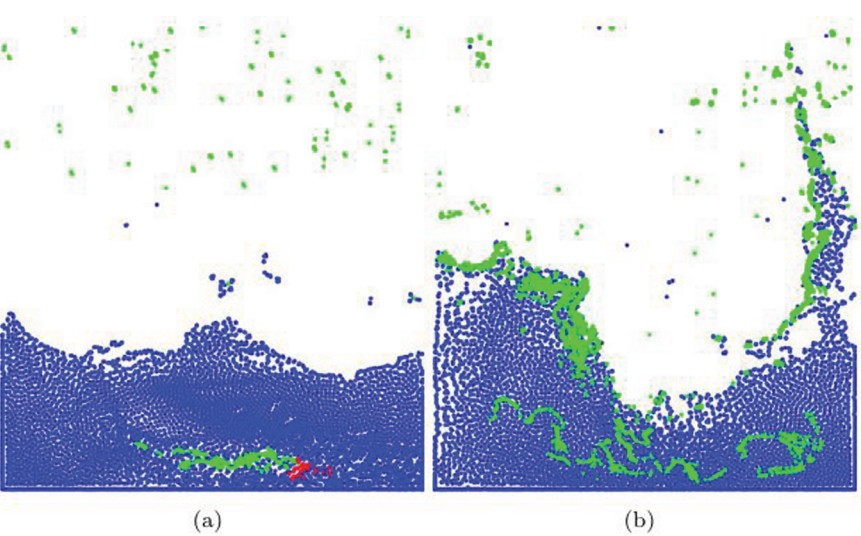

**Fig 19. Diffuse particles expressed by mouse interaction. This scene was created by integrating our method with a secondary effects solver (blue particle: water particle, green particle: diffuse particle, red particle: particles activated due to mouse interaction).**

https://doi.org/10.1371/journal.pcbi.0311163.g019

the surface. Furthermore, to determine the inside and outside of implicit surfaces, a structure with negative and positive values needs to be created. For this purpose, a small numerical $\epsilon$ value is subtracted from the approximated level-set value. As a result, in regions without water particles, $\phi$ becomes negative due to $\epsilon$, while in regions with particles, it becomes positive. Fluid surfaces can exist regardless of the velocity direction. Therefore, in our method, we consider both positive and negative search regions using $\Omega^{\pm}$ to extract the isolines of the fluid (see Fig 20). As shown in Fig 20, it effectively represents not only the splashing movement of the liquid caused by turbulent motion but also the scattering features.

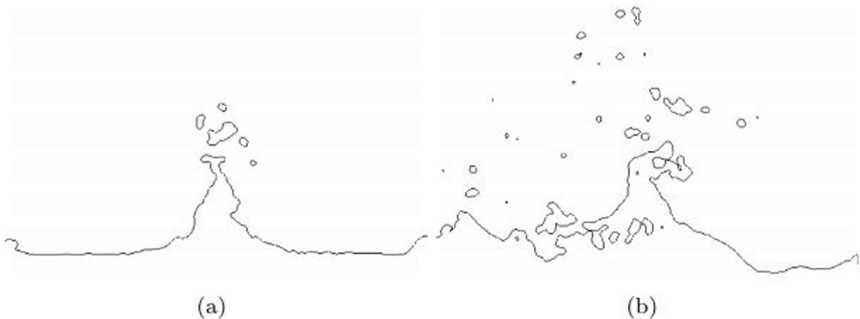

**Fig 20. Reconstructing fluid surfaces by integrating our method into an approximate scalar field.**

https://doi.org/10.1371/journal.pcbi.0311163.g020

**Collision handling in strand-based hair simulations.** Our method is applicable not only to particle-based fluids but also to hair/fur simulations composed in strand form. In hair simulations where the number of strands and particles is relatively high, collision handling is one of the bottleneck processes. In this section, we apply the adaptive disk sector to self-collision detection and analyze the results.

To compute the candidate primitive buffer required for rough collision detection, we collect edges composed of particles within the adaptive disk sector, $\Omega$. This enables us to calculate detailed collision detection and response. For the collision handling solver, we apply the Point-Point and Edge-Edge contacts from Chris Lewin's predictive contacts technique [46].

We have two particles with a radius $r$, each having positions $p_a$, $p_b$, and displacements $d_a$, $d_b$, all located within the adaptive disk sector ($p_a, p_b \in \Omega$). Based on these values, we can easily calculate whether particles are within the collision radius $r_{sc}$. We find the closest distance between $(p_a, p_a + d_a)$ and $(p_b, p_b + d_b)$, and if this distance is less than $r_{sc} + 2r$, a contact point is generated. If there is a high probability of collision, we need to create constraint conditions to prevent penetration. These constraints ensure that a plane with the normal $\hat{\mathbf{n}}$ does not penetrate the particle and they are described in Eqs 21a–21c. If the constraint conditions are satisfied, the particles will be positioned on the correct side relative to each other.

$$C(p_a, p_b) = \hat{\mathbf{n}} \cdot (p_a - p_b) - 2r \geq 0 \tag{21a}$$

$$\frac{\partial C}{\partial p_a} = \hat{\mathbf{n}} \tag{21b}$$

$$\frac{\partial C}{\partial p_b} = -\hat{\mathbf{n}} \tag{21c}$$

To compute Edge-Edge collision, information about the two target edges, $(p_a, p_b)$ and $(p_c, p_d)$, along with their displacements $(d_a, d_b)$ and $(d_c, d_d)$, is required ($p_a, p_b, p_c, p_d \in \Omega$). Then the closest points on the two edges, along with the interpolation parameters $\alpha$ and $\beta$ for each edge, are found ($p_\alpha = lerp(p_a, p_b, \alpha)$, $p_\beta = lerp(p_c, p_d, \beta)$). Then, the displacement of these points is calculated: $d_\alpha = lerp(d_a, d_b, \alpha)$, $d_\beta = lerp(d_c, d_d, \beta)$. This displacement is projected onto the discrete separation vector hatmathbf, and the separation and displacement speed are compared to the radius to determine if the edges are close.

To identify Edge-Edge contacts, constraints need to be calculated, and this is handled similarly to the Point-Point case. With constraints associated points, apart from the slightly offset

virtual points along each edge, named $p_\alpha$ and $p_\beta$, the process is the same as in the Point-Point contacts. Therefore, to identify contacts, topology, along with $\hat{\mathbf{n}}, \alpha$, and $\beta$, needs to be stored. Unlike the Point-Point case, this contact has only one extra layer with directionality, making the projection process relatively straightforward (see Eqs 22a–22e). For a more detailed explanation, it is recommended to refer to previously published relevant paper [46]. The crucial aspect of this process is to locate the necessary primitive sets for collision handling within $\Omega$.

$$C\left(p_a, p_b, p_c, p_d\right) = \left(p_\alpha - p_\beta\right) \cdot \hat{\mathbf{n}} - 2r \geq 0 \tag{22a}$$

$$\frac{\partial C}{\partial p_a} = -\left(1 - s\right)\hat{\mathbf{n}} \tag{22b}$$

$$\frac{\partial C}{\partial p_b} = -s\hat{\mathbf{n}} \tag{22c}$$

$$\frac{\partial C}{\partial p_c} = \left(1 - t\right)\hat{\mathbf{n}} \tag{22d}$$

$$\frac{\partial C}{\partial p_d} = t\hat{\mathbf{n}} \tag{22e}$$

Fig 21 shows the results obtained by integrating our method into the self-collision solver. As evident in the results, the collision detection was performed stably without any missing collisions. The collision response was conducted for bidirectional primitive edge pairs within the adaptive disk sector, ensuring stable collision handling without unstable forces.

**Turbulent flows in particle simulations.** Recently, Kim et al. proposed a method for calculating rotational momentum in particle-based fluids [47]. This method represents turbulent flow by calculating torque that expresses rotational momentum from neighbor particles. We compute the neighbor particle set $p_j$ of water particles within the adaptive disk sector $\Omega$. Rotational momentum is generated by the torque ($\tau = \mathbf{r} \times \mathbf{f}$), which is the result of the difference in position between the point where the force ($\mathbf{f}$) acts on the object and the object's center

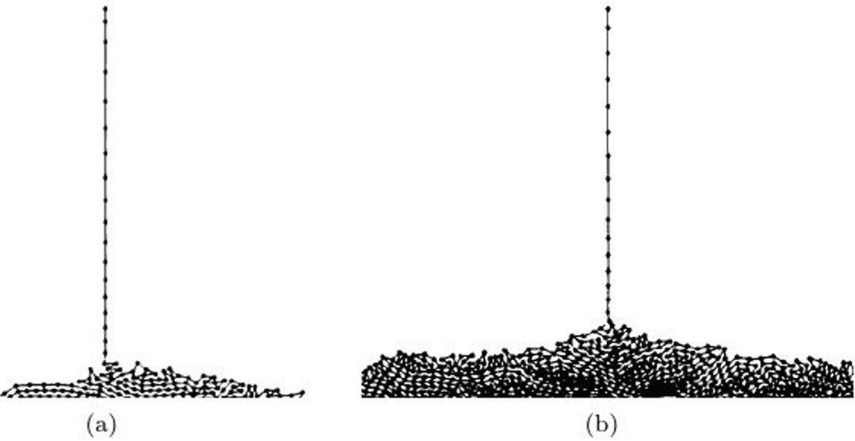

**Fig 21. Handling self-collision by integrating our method into collision solver.**

https://doi.org/10.1371/journal.pcbi.0311163.g021

of mass **r**. However, linear momentum has been considered without angular momentum in the case of a water particle, as the force acting on it is applied at the particle's position. In this paper, the torque on a given particle due to the force applied by a neighboring particle $j$ ($j \in \Omega$) is computed, and this torque is calculated as Eq 23.

$$\tau\left(\mathbf{x}\right) = \sum_{j \in \Omega} \left(\left(\mathbf{x} - \mathbf{x}_j\right) \times \mathbf{f}_j\right) W\left(\mathbf{x} - \mathbf{x}_j, h\right) \tag{23}$$

where $\tau$ represents the torque applied to the particle, $\mathbf{x}$ and $\mathbf{x}_j$ are the positions of the target particle and its neighboring particle, and W is the radially symmetric weighting function. The torque acting on the water particle is computed using Eq 23, and based on this, the angular momentum of the water particle is updated over time by integrating it as described in Eq 24.

$$\mathbf{L}_i^{n+1} = \mathbf{L}_i^n + \tau_i^n \Delta t \tag{24}$$

where $\mathbf{L}_i$ represents the angular momentum of the particle, and $\Delta t$ is the time step. The angular velocity of the particle is calculated using Eq 25.

$$\omega_i = I_i^{-1} \mathbf{L}_i^{n+1} \tag{25}$$

where $\omega_i$ is the angular velocity of the particle, and $I_i$ is the scalar inertia moment of the particle. In this paper, $I_i$ is set as $km_i$, where $m$ is the mass, and $k$ is $\frac{1}{30}$.

The previously calculated angular momentum of the particle affects the momentum of the particle, causing it to rotate. To calculate the change in linear momentum of the particle due to angular momentum, Eq 26 is used to compute the relative angular velocity between the target particle $i$ and neighbor particle $j$.

$$\omega_i^r = \frac{\left(\mathbf{x} - \mathbf{x}_j\right) \times \left(\mathbf{v} - \mathbf{v}_j\right)}{\left|\mathbf{x} - \mathbf{x}_j\right|^2} \tag{26}$$

where $\omega_i^r$ represents the relative angular velocity between the target particle $i$ and the surrounding particles. $\mathbf{x}$ and $\mathbf{x}_j$ represent the positions of the given particle and its neighbor particle, respectively, while $\mathbf{v}$ denotes the velocity of the particle and $j$ represents particles that exist within the adaptive disk sector ($j \in \Omega$). To incorporate the influence of particle angular velocity on linear momentum, the particle's velocity is updated using Eq 27.

$$\mathbf{v}_i^{n+1} = \mathbf{v}_i^* + \alpha \sum_{j \in \Omega} \left(\omega_i - \omega_j\right) \times \left(\mathbf{x} - \mathbf{x}_j\right) W\left(\mathbf{x} - \mathbf{x}_j, h\right) \tag{27}$$

where $= \mathbf{v}_i^*$ is the velocity of the particle before angular motion is applied. $\alpha$ is the particle's angular momentum transfer constant, and in this paper, it is set to 10. The goal of Eq 27 is to transfer the particle's angular velocity ($\omega_i$) to its linear momentum.

Fig 22 presents the results of integrating our method with the previous technique [47] and performing advection based on angular momentum. When comparing the velocity calculated in the previous method based on an isotropic kernel and using the weights of neighbor particles, our method, which employs an adaptive disk sector, generates results that are nearly identical.

Fig 23 visualizes the particle trajectories of our method and the previous method. The positions and velocities of the particles were randomly set, and then each method was used to

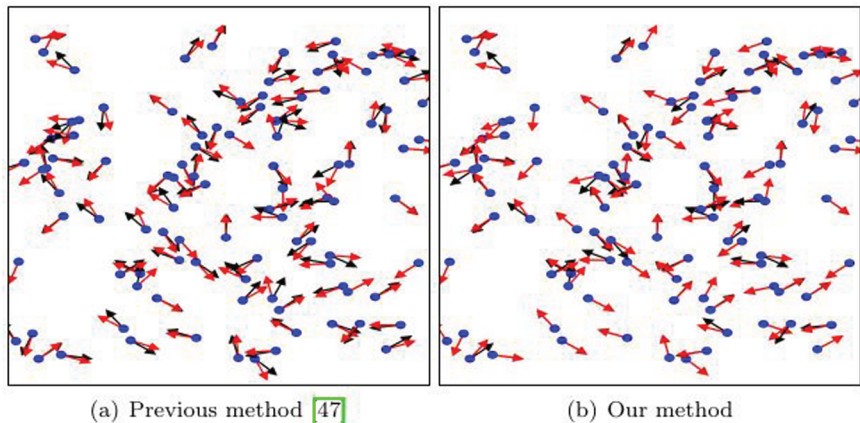

(a) Previous method [47]  (b) Our method

**Fig 22. Turbulent flow by in integrating our method into angular momentom of SPH particles (blue dot: particle position, black arrow: initial velocity, red arrow: velocity after angular momentum advection).**

https://doi.org/10.1371/journal.pcbi.0311163.g022

advect the particles. The resulting trajectories were visualized using lines. Fig 23a illustrates the results without turbulent flow, and the particle advection was modeled using the kernel form commonly used in Smoothed Particle Hydrodynamics (SPH) [48]. The initial random velocities lead to movement, and the isotropic spreading is evident due to the radially symmetric weighting kernel. This process demonstrates the representation of Naïve SPH motion without incorporating turbulent flow or vortex effects.

Fig 23b shows the results of applying the advection technique modeled based on angular momentum to particle-based fluids [47]. The results demonstrate that rotational momentum is well maintained and represented compared to the outcomes in Fig 23a. However, this technique can lead to instability in the simulation due to the continuous increase in angular momentum, resulting in occasional noise. On the other hand, when integrating this technique with our method and performing advection, the distinctive features are clearly evident in the trajectory of particles (see Fig 23c). This is because angular momentum is calculated using the particles within the adaptive disk sector $\Omega$, which is generated in the direction of the given particle's velocity. Angular momentum is generated and preserved in the direction of the velocity, and it gradually diminishes as particles move outward. These results demonstrate that, similar to a typical radially symmetric weighting function, momentum dissipates in regions with low density, while an anisotropic kernel effectively captures turbulent flow in the direction of velocity. These characteristics demonstrate that the issues of noise and over-preserved angular momentum, raised as problems in previous methods, have been addressed by using an adaptive disk sector.

## Discussion

In this paper, we proposed the adaptive disk sector technique and demonstrated its superiority and versatility by integrating it into various methods, showing the results of our experiments. We also demonstrated computational efficiency by adjusting the sampling count for the NNP and weighting based on the particle's velocity. However, when performing isoline tracking, the accuracy may be lower compared to directly calculated anisotropic kernel-based level-set. In this process, $\Omega^{\pm}$ was used to consider both directions of velocity, but since this area is not computed through numerical analysis, accuracy may be compromised. Therefore,

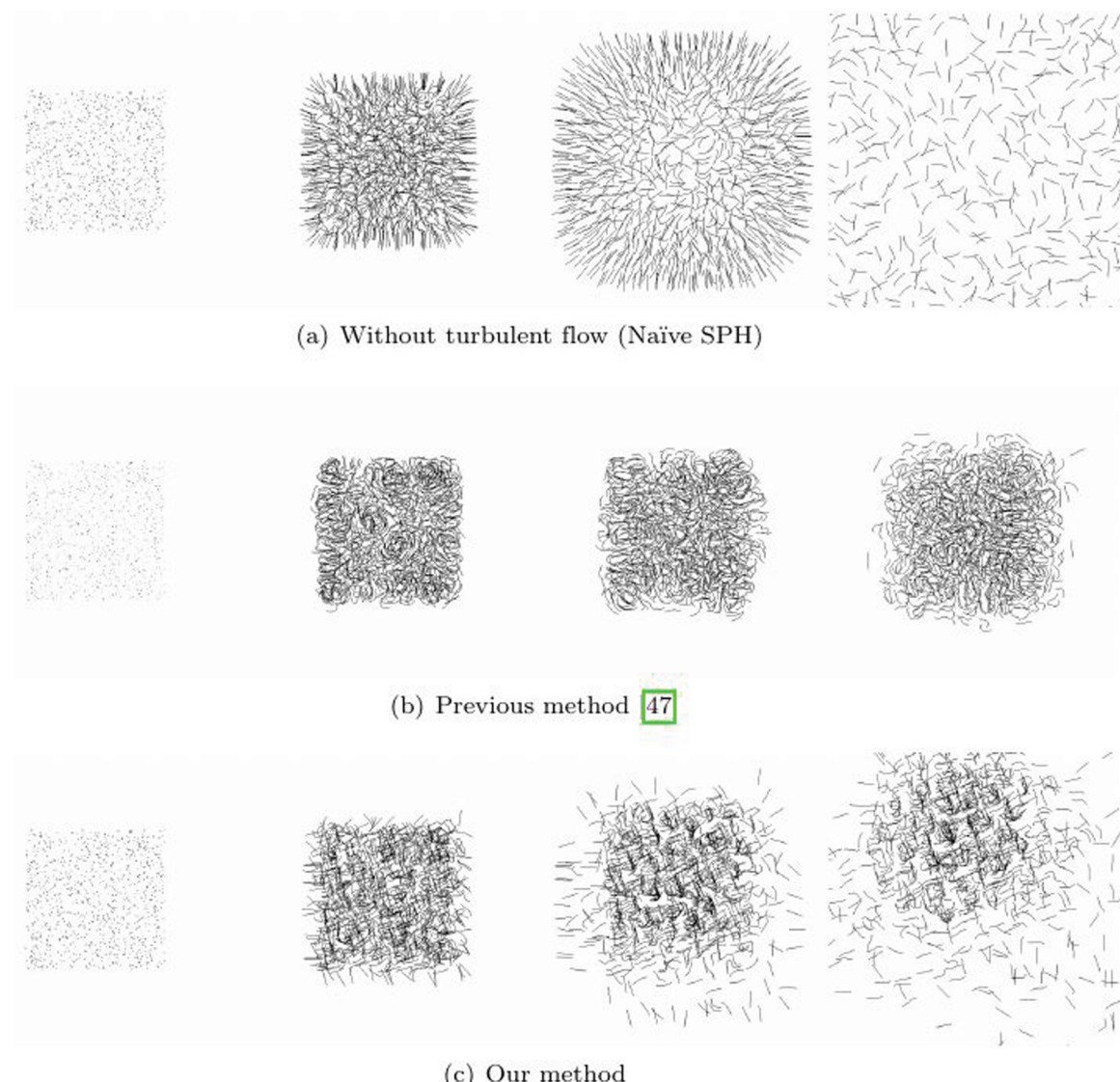

(a) Without turbulent flow (Naïve SPH)

(b) Previous method [47]

(c) Our method

**Fig 23. Comparison results of particle trajectories between our method and previous methods.**

https://doi.org/10.1371/journal.pcbi.0311163.g023

to perform surface reconstruction based on anisotropic kernels using the proposed method, further algorithm improvements are necessary.

## Conclusion and future work

In this paper, a novel framework was proposed for efficiently computing NNP in movable particle-based systems such as SPH or hair particles by dynamically adapting the disk sector. In this study, we proposed a method to rapidly define a disk sector-shaped search range for moving particles based solely on their velocity, without relying on explicit data structures like hash tables or $K$-d trees. Additionally, to update the directional search range, we introduced a kernel that maps the particle's velocity to the angle of the search range, enabling stable updates to the disk sector range. In contrast to previous methods that perform NNP checks within

a fixed range, our study introduces a novel framework that dynamically adjusts the range of the adaptive disk sector based on the particle's velocity. This framework aims to reduce the number of particles needed for NNP checks. The proposed framework was applied to various applications, demonstrating its scalability and effectiveness. The adaptive disk sector was defined such that the disk is positioned on the particle conducting the NNP checks. The angle and length of the disk were dynamically adjusted based on the particle's velocity. When the particle's velocity is slow, the length of the disk sector was set to be short, and the angle was set to be wide. On the other hand, when the velocity is fast, the opposite configuration was used, with a longer length and a narrower angle. We observed a significant improvement in the performance of NNP checks in a 2D particle-based system, with performance gains ranging from 2 to 20 times.

We conducted experiments in several scenes to validate the effectiveness of our proposed method. These experiments include: 1) Particle placement through random sampling, 2) particle placement with random sampling at various densities (see Figs 8, 9, and 10), 3) particle placement using SPH simulation that exhibits regular distribution, not based on random sampling, and 4) particle placement using hair simulation that demonstrates a more irregular distribution than SPH. In this paper presents our method's robust applicability across a wider variety of environments through the four mentioned scenarios in the 'Applications' section.

In performance metrics, the computation time using NNP is crucial, and we have tested its capability to stably perform collision handling, surface tracking, and secondary effects across various applications. Our method does not merely rely on discretized space to find NNPs but also considers the velocity of the target particle, during which we compared the number of NNPs (see Fig 9). Because our method takes into account the moving particle's velocity, it has shown relatively stable results compared to other approaches. Stability means maintaining an appropriate number of NNPs that is neither too few nor too many, in accordance with the target particle's velocity.

The method proposed in this paper focuses on efficiently calculating NNP, and the best way to demonstrate its performance is by measuring the trade-off between quality and efficiency. The ellipsoidal-kernel-based particle-based fluids and diffuse particles, shown in various applications, are modeled based on the momentum of neighboring particles. While they appear similar in terms of visual quality, our method shows improved efficiency by finding NNPs more effectively. This is demonstrated across various scenarios in the Results section. The diffuse particles and surface tracking sections reveal visually well-represented outcomes. Additionally, collision detection was performed effectively without any missed collisions. Since our approach enhances computational performance by reducing the number of candidate particles when calculating NNPs, it shows no significant difference in terms of quality. Nevertheless, even in the final application, turbulent flow, we observed that turbulence was generated and maintained well, with no noticeable noise compared to previous methods (see Fig 23).

However, despite these improvements, our method still has a few drawbacks. Since the algorithm was designed based on a 2D framework, it needs to be extended to 3D to be applicable in a wider range of fields. Once the extension to 3D is completed, we anticipate that the algorithm can be applied not only to particle-based simulations but also to rendering techniques for computing photon energy in indirect illumination. Furthermore, during the isoline tracking process, surfaces are not limited to the velocity direction alone, which occasionally resulted in the loss of small splash effects and similar details. To mitigate such issues, it may be necessary to expand the algorithm by automatically controlling not only the velocity but also the orientation of the disk as needed. In the future, we plan to conduct research

aimed at applying the algorithm to various applications that require solving NNP$^{\star}$ problems by addressing the limitations mentioned earlier.

In this study, we focused on how to efficiently generate and update disk sectors in response to particle movements. Although many existing methods model shapes like disks or cones, we chose to use a relatively simple implicit function form to perform updates efficiently. To initially filter particles with minimal computation, we utilized a sphere equation. As far as we know, our paper is the first to propose a method for generating and updating disk sectors to construct NNPs. While there are still many challenges to address, we plan to further optimize and improve this algorithm in the future.

## Author contributions

**Funding acquisition:** Jung Lee.

**Methodology:** Jong-Hyun Kim.

**Software:** Jong-Hyun Kim.

**Supervision:** Jong-Hyun Kim.

**Visualization:** Jung Lee.

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
