## [Decision Letter · Decision Letter 0]

16 Aug 2024

PONE-D-24-28062Scalable and Rapid Nearest Neighbor Particle Search Using Adaptive Disk SectorPLOS ONE

Dear Dr. Lee,

Thank you for submitting your manuscript to PLOS ONE. After careful consideration, we feel that it has merit but does not fully meet PLOS ONE’s publication criteria as it currently stands. Therefore, we invite you to submit a revised version of the manuscript that addresses the points raised during the review process.

We look forward to receiving your revised manuscript.

Kind regards,

Shaofeng Xu

Academic Editor

PLOS ONE

Reviewers' comments:

Reviewer's Responses to Questions

**Comments to the Author**

1. Is the manuscript technically sound, and do the data support the conclusions?

Reviewer #1: Yes

Reviewer #2: Yes

Reviewer #3: Yes

2. Has the statistical analysis been performed appropriately and rigorously? 

Reviewer #1: No

Reviewer #2: Yes

Reviewer #3: Yes

3. Have the authors made all data underlying the findings in their manuscript fully available?

Reviewer #1: No

Reviewer #2: No

Reviewer #3: No

4. Is the manuscript presented in an intelligible fashion and written in standard English?

Reviewer #1: Yes

Reviewer #2: Yes

Reviewer #3: Yes

5. Review Comments to the Author

Reviewer #1: The paper proposed Scalable and Rapid Nearest Neighbor Particle Search Using Adaptive Disk Sector. The paper require major revision:

1. The contribution should be listed correctly. The context of the contribution is vague.

2. Any justification of why Equation (1) must be considered in this case?

3. Kindly justify the convergence of Equation (7) in relation to your proposed work.

4. As for Fig 5, the author uses brute force which is the worst way to find the optimal solution. Any justification on this?

5. Fig 11 should be explained,

6. As for Table 2, please conduct Friedman test to validate the superiroty of your proposed work. You may refer/cite the following work

(a) Supervised learning perspective in logic mining

(b) Novel logic mining incorporating log linear approach

Please conduct sufficient hypothesis testing to validate your work.

7. The author should conduct convergence analysis to validate the proposed work.

8. As for application, please justify how do you choose the applications. Any citation to support your justifcation?

9. Qualitative analysis should be conducted in all applications

10. The conclusion must be mapped with the contributions.

Reviewer #2: Comments to the Author

Feedback:

Please find to follow my comments on the paper entitled: Scalable and Rapid Nearest Neighbor Particle Search Using Adaptive Disk Sector.

The authors propose a novel data structure for accelerating NNP searches in 2D space by dynamically altering disk sectors based on the positions and velocities of particles. The introduction is quite generic, please detail in a wider way to extend their main contribution to other research fields . There is no in-depth discussion about the advantages of the technique proposed in comparison between state of the art . Please add citations for this section.

Expression a¿b=c is unclear Line 490

Cases for Fig 2. (A), (B), and (C) , should use lowercase letters.

use the symbol δ instead of delta to define weight in line 288 from Eq(4).

Axis from Fig 4 needs to be detailed or described.

Fig 5a and Fig 5b are not explained.

What does ´´Res´´ mean from Fig 8?; Do you mean Resolution?

From Fig 9 (X-axis : frame #, Y-axis : number of NNPs) I recommend put these in the corresponding axis of each Figures.

There are K-d trees in the manuscript that are not italicized.

It would be gratefully appreciated if the author can share the pseudocode and data set from the experimental set.

Reviewer #3: The paper is well written and clearly illustrated. Scalability could be a problem. It looks like the experiments have been done only on 10,000 particles and there are too many assumptions about the configuration of the particle at the point of collision. This can reduce the practicality of the solution.

6. PLOS authors have the option to publish the peer review history of their article (what does this mean?). If published, this will include your full peer review and any attached files.

Reviewer #1: No

Reviewer #2: No

Reviewer #3: No

---

## [Author Response · Author response to Decision Letter 1]

28 Aug 2024

I have attached the following file 'Respond to Reviewers.docx'

---

## [Decision Letter · Decision Letter 1]

16 Sep 2024

Scalable and Rapid Nearest Neighbor Particle Search Using Adaptive Disk Sector

PONE-D-24-28062R1

Dear Dr. Lee,

We’re pleased to inform you that your manuscript has been judged scientifically suitable for publication and will be formally accepted for publication once it meets all outstanding technical requirements.

Kind regards,

Shaofeng Xu

Academic Editor

PLOS ONE

Additional Editor Comments (optional):

Reviewers' comments:

Reviewer's Responses to Questions

**Comments to the Author**

1. If the authors have adequately addressed your comments raised in a previous round of review and you feel that this manuscript is now acceptable for publication, you may indicate that here to bypass the “Comments to the Author” section, enter your conflict of interest statement in the “Confidential to Editor” section, and submit your "Accept" recommendation.

Reviewer #1: All comments have been addressed

Reviewer #2: All comments have been addressed

2. Is the manuscript technically sound, and do the data support the conclusions?

Reviewer #1: Partly

Reviewer #2: Yes

3. Has the statistical analysis been performed appropriately and rigorously? 

Reviewer #1: No

Reviewer #2: Yes

4. Have the authors made all data underlying the findings in their manuscript fully available?

Reviewer #1: No

Reviewer #2: Yes

5. Is the manuscript presented in an intelligible fashion and written in standard English?

Reviewer #1: Yes

Reviewer #2: Yes

6. Review Comments to the Author

Reviewer #1: The author has addressed almost all my comments. Since there is no comment left, i hope this paper will benefit the future readers

Reviewer #2: The authors have improved the manuscript as suggested by this reviewer.

The introduction was quite generic, then the authors changed the presentation. The contribution of the paper is clear a now is detailed in a wider way to be extented to other research field.

7. PLOS authors have the option to publish the peer review history of their article (what does this mean?). If published, this will include your full peer review and any attached files.

Reviewer #1: No

Reviewer #2: No

---

## [Editor Report · Acceptance letter]

PONE-D-24-28062R1

PLOS ONE

Dear Dr. Lee,

I'm pleased to inform you that your manuscript has been deemed suitable for publication in PLOS ONE. Congratulations! Your manuscript is now being handed over to our production team.

Kind regards,

on behalf of

Dr. Shaofeng Xu

Academic Editor

PLOS ONE